# Niacin ameliorates ulcerative colitis via prostaglandin D$_2$-mediated D prostanoid receptor 1 activation

Juanjuan Li[1,2,†], Deping Kong[2,†], Qi Wang[1,†], Wei Wu[1,†], Yanping Tang[1], Tingting Bai[1], Liang Guo[3,4], Lumin Wei[1,2], Qianqian Zhang[2], Yu Yu[2], Yuting Qian[1], Shengkai Zuo[2], Guizhu Liu[2], Qian Liu[2], Sheng Wu[1], Yi Zang[1], Qian Zhu[2], Daile Jia[2], Yuanyang Wang[5], Weiyan Yao[1], Yong Ji[6], Huiyong Yin[2], Masataka Nakamura[7], Michael Lazarus[8], Richard M Breyer[9,10], Lifu Wang[1,*] & Ying Yu[2,5,**]

## Abstract

Niacin, as an antidyslipidemic drug, elicits a strong flushing response by release of prostaglandin (PG) D$_2$. However, whether niacin is beneficial for inflammatory bowel disease (IBD) remains unclear. Here, we observed niacin administration-enhanced PGD$_2$ production in colon tissues in dextran sulfate sodium (DSS)-challenged mice, and protected mice against DSS or 2,4,6-trinitrobenzene sulfonic acid (TNBS)-induced colitis in D prostanoid receptor 1 (DP1)-dependent manner. Specific ablation of DP1 receptor in vascular endothelial cells, colonic epithelium, and myeloid cells augmented DSS/TNBS-induced colitis in mice through increasing vascular permeability, promoting apoptosis of epithelial cells, and stimulating pro-inflammatory cytokine secretion of macrophages, respectively. Niacin treatment improved vascular permeability, reduced apoptotic epithelial cells, promoted epithelial cell update, and suppressed pro-inflammatory gene expression of macrophages. Moreover, treatment with niacin-containing retention enema effectively promoted UC clinical remission and mucosal healing in patients with moderately active disease. Therefore, niacin displayed multiple beneficial effects on DSS/TNBS-induced colitis in mice by activation of PGD$_2$/DP1 axis. The potential efficacy of niacin in management of IBD warrants further investigation.

**Keywords** DP1 receptor; niacin; prostaglandin; retention enema; ulcerative colitis

**Subject Categories** Digestive System; Immunology

## Introduction

Ulcerative colitis (UC) is a chronic inflammatory bowel disease (IBD) characterized by recurrent episodes of active disease, which commonly affects the colon, the rectum, or both simultaneously. Histologically, it displays chronic inflammatory alterations limited to the mucosa and submucosa with cryptitis and crypt abscesses (Danese & Fiocchi, 2011). Despite UC-related mortality being low, its morbidity remains high and 10–20% of affected individuals undergo colectomy. Although the UC etiology is largely unknown, accumulated evidence supports an interaction between genetic predisposition and microbial/environmental factors that trigger pro-colitogenic perturbations of the host–commensal relationship and an aberrant mucosal immune response (Khor et al, 2011). Genome-wide association studies (GWAS) have identified 47 genetic susceptibility loci for UC, 28 of which are shared between Crohn's disease (CD) and UC (Franke et al, 2010; Anderson et al, 2011). Indeed, these risk loci implicated in IBD are involved in different key signal pathways which are essential for intestinal homeostasis, such as epithelial restitution, barrier function, innate and adaptive immune regulation, microbial defense, cellular stress, and metabolism (Khor et al, 2011). Moreover, vascular injury including dilated vessels and

1   Department of Gastroenterology, Ruijin Hospital affiliated to Shanghai Jiao Tong University School of Medicine, Shanghai, China
2   Key Laboratory of Food Safety Research, Institute for Nutritional Sciences, Shanghai Institutes for Biological Sciences, Chinese Academy of Sciences, Shanghai, China
3   Department of Breast Surgery, Breast Cancer Institute, Fudan University Shanghai Cancer Center, Shanghai, China
4   Department of Oncology, Shanghai Medical College, Fudan University, Shanghai, China
5   Department of Pharmacology, School of Basic Medical Sciences, Tianjin Medical University, Tianjin, China
6   The Key Laboratory of Cardiovascular Disease and Molecular Intervention, Atherosclerosis Research Centre, Nanjing Medical University, Nanjing, Jiangsu, China
7   Human Gene Sciences Center, Tokyo Medical and Dental University, Bunkyo-ku, Tokyo, Japan
8   International Institute for Integrative Sleep Medicine (WPI-IIIS), University of Tsukuba, Tsukuba City, Ibaraki, Japan
9   Department of Veterans Affairs, Tennessee Valley Health Authority, Nashville, TN, USA
10  Department of Medicine, Vanderbilt University Medical Center, Nashville, TN, USA
    *Corresponding author. Tel: +86 021 64370045 665290; Fax: +86 021 66295266; E-mail: lifuwang@sjtu.edu.cn
    **Corresponding author. Tel: +86 21 54920970; E-mails: yuying@sibs.ac.cn; yuying@tmu.edu.cn
    †These authors contributed equally to this work

increased vascular permeability also contributes to the inflammatory disorder of colonic mucosa in UC patients (Deng *et al*, 2013).

Niacin (nicotinic acid) is also known as vitamin B3 and serves as a precursor for coenzymes such as nicotinamide adenine dinucleotide (NAD) and nicotinamide adenine dinucleotide phosphate (NADP), which are essential for living cells. Niacin has been used for more than five decades to treat dyslipidemia, because it reduces low-density lipoprotein cholesterol (LDLc), very low-density lipoprotein cholesterol (VLDLc), and triglycerides (TGs), and elevates high-density lipoprotein cholesterol (HDLc) (Song & FitzGerald, 2013). The orphan G-protein-coupled receptor GPR109A, also known as hydroxy-carboxylic acid 2 (HCA2) in mice, and as HM74A in humans, can be activated by niacin (Wise *et al*, 2003). The beneficial effect of niacin on free fatty acid and TGs is mediated by GPR109A suppression of lipolysis; however, the effects on HDLc and LDLc are not mediated by the GPR109A receptor (Bodor & Offermanns, 2008). GPR109A expression is markedly upregulated in macrophages upon inflammatory stimulation (Feingold *et al*, 2014). Moreover, emerging evidence demonstrated that niacin displays multiple anti-inflammatory properties through GPR109A receptor activation (Holzhauser *et al*, 2011; Digby *et al*, 2012; Godin *et al*, 2012; Zandi-Nejad *et al*, 2013; Zhou *et al*, 2014). Thus, the potential therapeutic efficacy of niacin on patients with UC warrants further clinical investigation.

One unpleasant side effect caused by niacin is cutaneous flushing. Niacin stimulates prostaglandin $D_2$ ($PGD_2$) release in both mice and humans (Hanson *et al*, 2010; Song & FitzGerald, 2013), which plays a central role in the niacin-induced flushing. Low-dose aspirin could depress niacin-evoked $PGD_2$ release and reduce the associated flushing (Cefali *et al*, 2007; Song & FitzGerald, 2013). $PGD_2$ promotes the niacin-evoked flushing through its specific D prostanoid receptor 1 (DP1). Blockade of DP1 receptor completely inhibits niacin-induced vasodilation in mice and humans without affecting its effects on lipid metabolism (Cheng *et al*, 2006; Paolini *et al*, 2008; Maccubbin *et al*, 2009). In addition, $PGD_2$ mediates active resolution of inflammation through DP1 receptor (Rajakariar *et al*, 2007; Kong *et al*, 2016). Interestingly, marked elevation of $PGD_2$ production was observed in inflamed colon tissues from both UC patients and experimental colitis murine models (Ajuebor *et al*, 2000; Vong *et al*, 2010), which is associated with long-term remission in humans (Vong *et al*, 2010). Yet, it remains to be determined whether niacin-mediated protection against UC depends on $PGD_2$ production.

In this study, we investigated the therapeutic effect of niacin on colitis both in mice and in patients with moderately active UC. We found that niacin shows anti-inflammatory and anti-apoptotic properties through downregulation of colonic inflammatory cytokine levels, suppression of vascular permeability, and inhibition of colonic epithelium apoptosis by activation of DP1 receptor in macrophages, endothelial cells, and colonic epithelium. Furthermore, treatment with retention enema containing niacin effectively promoted clinical remission and mucosal healing in patients with moderately active UC.

# Results

## Niacin boosts $PGD_2$ generation in mice

To explore whether niacin protects against inflammatory bowel diseases (IBDs) through releasing $PGD_2$, we first examined niacin-induced $PGD_2$ production in colon tissues and urinary secretion of $PGD_2$ metabolites- 11,15-Dioxo-9α-hydroxy-2,3,4,5-tetranor-prostan-1,20-dioic acid (tetranor PGDM) from DSS-induced colitis mouse model by using mass spectrometry analysis. Indeed, $PGD_2$ production in homogenized colons and urinary tetranor PGDM was markedly elevated by niacin administration in DSS-challenged mice in a dose-dependent manner (Fig 1A and B). In addition, niacin treatment induced $PGF_{2\alpha}$ product in colon tissues (Fig EV1A) and increased urinary metabolites of $PGE_2$, $PGI_2$, and $PGF_{2\alpha}$ (Fig EV1B) in DSS-challenged mice, indicating niacin may upregulate PG biosynthesis pathway. Accordingly, we observed niacin treatment upregulated cytosolic phospholipase $A_2$ ($cPLA_2$), COX-2, and hematopoietic PGD synthase (hPGDS) in peritoneal macrophages (Fig 1C–E). However, niacin had no markedly influence on specialized pro-resolving mediators (SPMs) in colon tissues from DSS-challenged mice, such as lipoxin (LX) A4, resolvin (Rv) E1 (Fig EV1C).

## Disruption of DP1 receptor deteriorates both DSS- and TNBS-induced colitis in mice

$PGD_2$ specifically binds and activates two distinct D prostanoid receptors DP1 and DP2. Next, we investigated the effects of $PGD_2$ receptor deficiency on development of DSS- or TNBS-induced colitis in mice. Interestingly, mice with global DP1 disruption (Fig 2A) lost over 12% more weight than wild-type (WT) controls (Fig 2B), and had significantly higher DAI than WT after DSS challenge (2.33 ± 0.33 vs. 0.42 ± 0.22, $P < 0.01$, Fig 2C). Accordingly, DP1 deletion augmented the severity of DSS-induced colitis in mice including reduction of colon length (Fig 2D), increase of epithelial cell lost, thickening of intestinal wall, enhanced infiltration of inflammatory cells in colon tissues (Fig 2E), and increase of overall mortality (Fig 2F). Likewise, $DP1^{-/-}$ mice were also more vulnerable to TNBS-induced colitis (Fig EV2). However, DP2 deficiency (Satoh *et al*, 2006) did not influence DSS-induced colitis in mice (Fig 2A–F). Thus, activation of DP1 receptor, not DP2, protects mice against DSS/TNBS-induced colitis.

## Niacin ameliorates DSS/TNBS-induced colitis in mice through DP1 receptor

Treatment with niacin promotes $PGD_2$ release in colon tissues (Fig 1A), and disruption of DP1 receptor worsens DSS/TNBS-induced colitis in mice (Figs 2 and EV2). We hypothesized niacin could improve clinical manifestation of colitis induced by DSS or TNBS. Indeed, administration of niacin (600 mg/kg by gavage, once a day, Fig 3A) markedly delayed body weight loss (7.94 ± 1.44% vs. 14.25 ± 1.03%, $P < 0.01$, Fig 3B), elevation of DAI (1.46 ± 0.14 vs. 2.58 ± 0.16, $P < 0.01$, Fig 3C), and shortening of colon length (6.26 ± 0.19 cm vs. 4.43 ± 0.27 cm, $P < 0.01$, Fig 3D and E) caused by DSS challenge in WT mice, and consequently reduced mortality (Fig 3F). In addition, niacin also ameliorated body weight loss and shortening of colon length caused by TNBS challenge in mice (Fig EV3). In contrast, these beneficial effects of niacin were not observed in DP1-deficient mice (Figs 3B–F and EV3).

We also examined the effect of niacin on oxidative stress and plasma lipids in mice. As shown in Appendix Fig S1, the levels of 8-isoprostane $PGF_{2\alpha}$ in colon tissues and urine were not altered

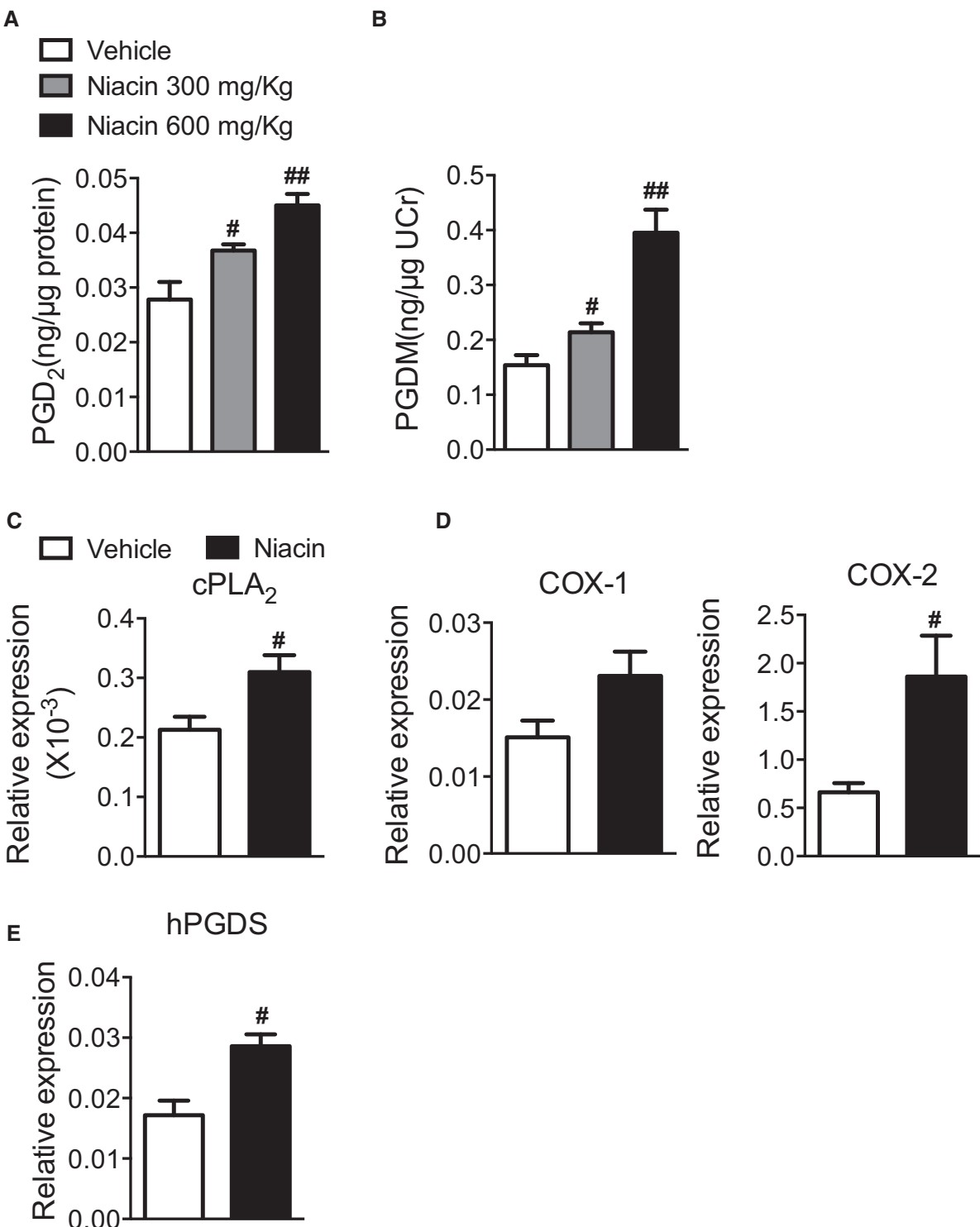

**Figure 1.  Niacin induces PGD$_2$ secretion in DSS-challenged mice.**

A    Mass spectrometry analysis of PGD$_2$ production in colons from niacin-treated mice after DSS challenge.
B    Mass spectrometry analysis of urinary PGD$_2$ metabolites (PGDM) from niacin-treated mice after DSS challenge. PGDM, 11,15-dioxo-9 alpha-hydroxy-2,3,4,5-tetranorprostan-1,20-dioic acid.
C–E   Real-time PCR analysis of cPLA$_2$, COX-1, COX-2, and hPGDS expression in peritoneal macrophage treated with niacin.

Data information: Data are shown as mean $\pm$ SEM. Data are representative of at least two independent experiments. Statistical significance was determined using unpaired Student's *t*-tests. (A) [#]$P < 0.05$, [##]$P < 0.01$ vs. vehicle; vehicle, $n = 6$; niacin 300 mg/kg, $n = 5$; niacin 600 mg/kg, $n = 7$. (B) [#]$P < 0.05$, [##]$P < 0.01$ vs. vehicle; $n = 6$. (C–E) [#]$P < 0.05$ vs. vehicle; $n = 4$.

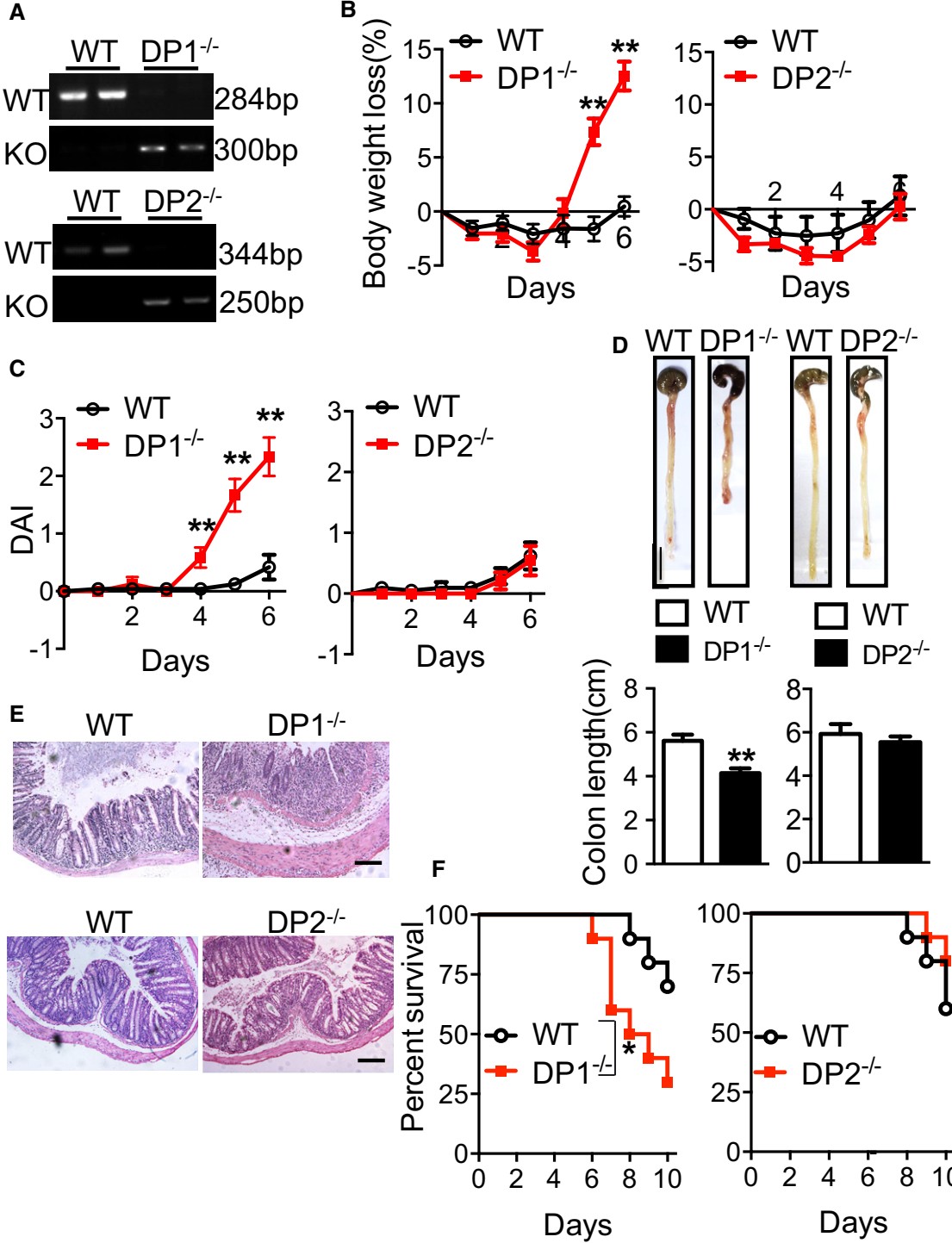

**Figure 2.  DP1 knockout augments DSS-induced colitis in mice.**

A       PCR analysis of tail genomic DNA from DP1$^{-/-}$, DP2$^{-/-}$, and WT mice.

B–D    Body weight loss (B) and disease activity index (C), and colon length (D) of DP1$^{-/-}$, DP2$^{-/-}$, and WT mice in response to DSS challenge. Scale bar: 1 cm.

E       H&E staining of histological sections in the distal colon from the mice administered with DSS for 6 days. Scale bars: 100 μm. Graphs represent overall histology score.

F       Survival rates of DSS-challenged DP1$^{-/-}$, DP2$^{-/-}$ mice, and WT controls.

Data information: Data are shown as mean ± SEM. Data are representative of three independent experiments. (B–D) Statistical significance was determined using unpaired Student's *t*-tests. **$P < 0.01$ compared with WT. Left panel: WT, $n = 8$; DP1$^{-/-}$, $n = 8$. Right panel: WT, $n = 7$; DP2$^{-/-}$, $n = 8$. (F) Survival rate was compared using the log-rank test. *$P < 0.05$, compared with WT; $n = 10$.

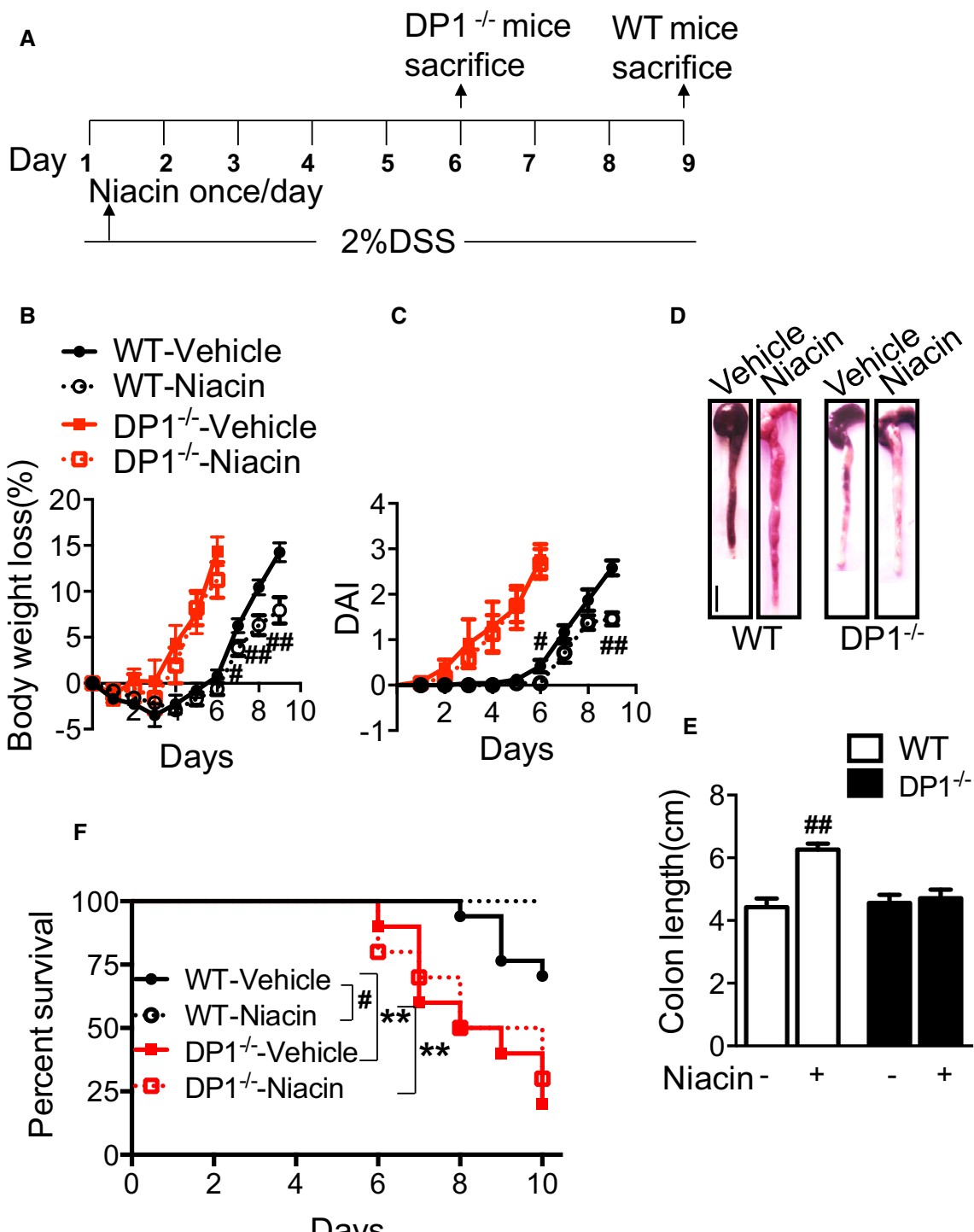

**Figure 3. Niacin protects mice from DSS-induced colitis.**

A    Protocol for niacin treatment on DSS-induced colitis in mice.
B, C    Effect of niacin treatment on body weight loss (B) and disease activity index (C) of DP1$^{-/-}$ and WT mice in response to DSS challenge.
D    Macroscopic appearance of colons from DSS-challenged mice after niacin treatment. Scale bar: 1 cm.
E    Effect of niacin treatment on colon length (centimeter) of DP1$^{-/-}$ and WT mice in response to DSS challenge.
F    Effect of niacin treatment on survival rates of DSS-challenged DP1$^{-/-}$ mice and WT controls.

Data information: Data are representative of two independent experiments. (B, C, E) Data are shown as mean ± SEM. Statistical analysis was performed using unpaired Student's *t*-test. #$P < 0.05$, ##$P < 0.01$ vs. vehicle; $n = 8$. (F) Survival rate was compared using the log-rank test. #$P < 0.05$, vs. vehicle, *$P < 0.05$, **$P < 0.01$ compared with WT; WT, $n = 17$; DP1$^{-/-}$, $n = 20$.

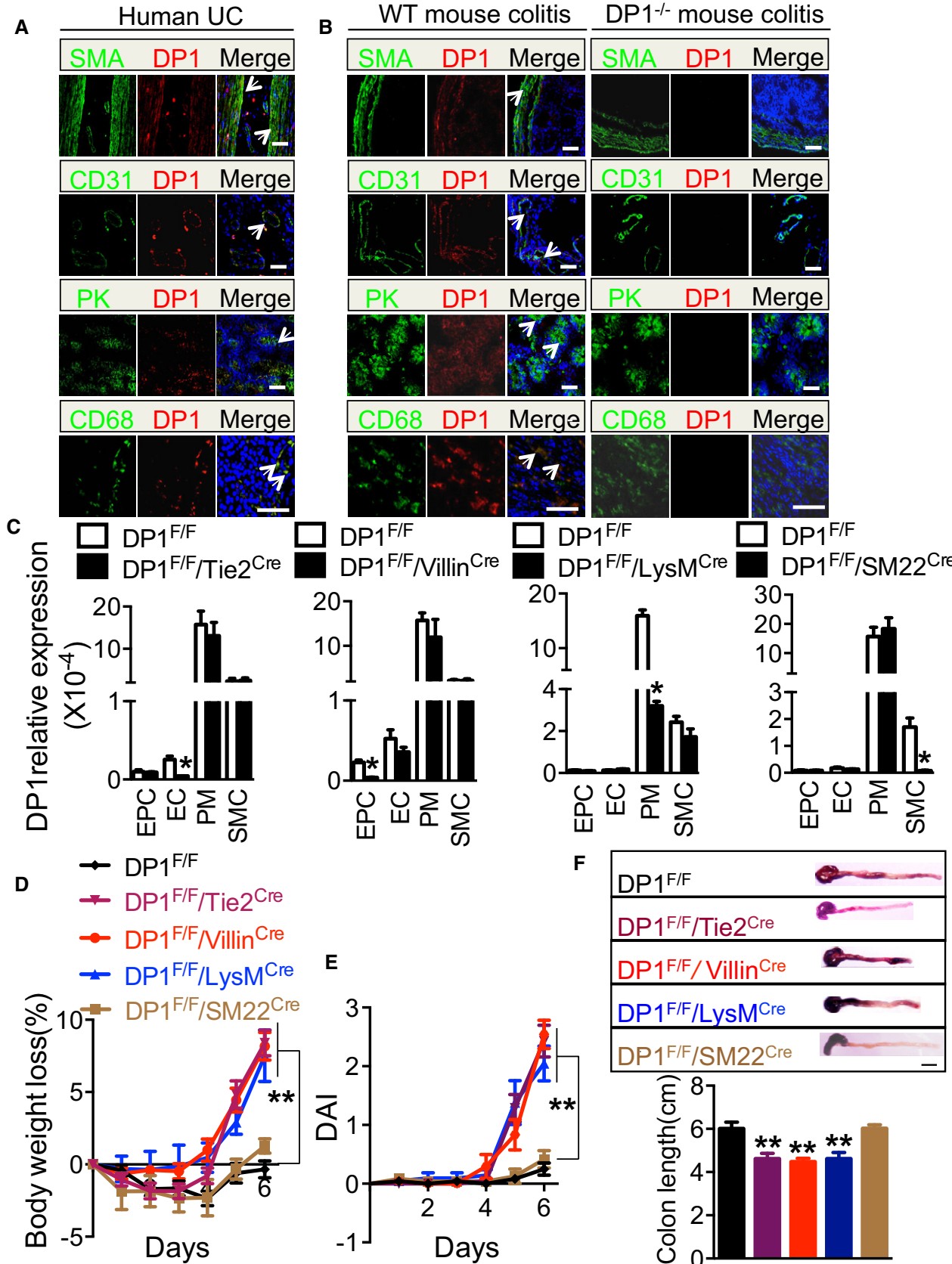

**Figure 4.**

(Appendix Fig S1A) by niacin treatment in mice, while plasma cholesterol and triglyceride were markedly decreased after niacin treatment (Appendix Fig S1B).

## DP1 deletion in vascular endothelial cells, epithelial cells, and myeloid cells, but not in smooth muscle cells, exacerbates the development of DSS/TNBS-induced colitis in mice

DP1 receptor is expressed in colon tissues from healthy subjects, but its downregulation was detected in patients with active colitis (Vong *et al*, 2010). To determine which cell types express DP1 receptor, we performed double immunofluorescence staining of DP1 and different cell type-specific markers in inflamed colon tissues from both UC patients and DSS-challenged mice. Sections were stained for DP1 along with a α-actin, a smooth muscle cell (SMA)-specific marker, CD31, an endothelial cell marker, pan-keratin, an epithelial cell marker, or CD68, a macrophage marker. Doubly stained cells were observed among all these marker positive cells (Fig 4A and B), indicating DP1 was expressed variably in multiple cell types including smooth muscle cells, vascular endothelial cells, colonic epithelial cells, and macrophages.

To identify the cell types in which loss of DP1 function exaggerated the murine colitis, DP1$^{Flox}$/$^{Flox}$ (DP1$^{F/F}$) mice were crossed with Tie2$^{Cre}$, Villin$^{Cre}$, LysM$^{Cre}$, or SM22$^{Cre}$ transgenic mice to generate vascular endothelial cell, colonic epithelial cell, macrophage, or smooth muscle cell-specific DP1-deficient mice, respectively (Fig 4C). Since Tie2 kinase is also expressed in hematopoietic progenitors (Zhou *et al*, 2009), DP1 downregulation (~22%) in colonic macrophages was detected in DP1$^{F/F}$/Tie2$^{Cre}$ mice (data not shown). As shown in Figs 4D–F and EV4A, DP1$^{F/F}$/Tie2$^{Cre}$, DP1$^{F/F}$/Villin$^{Cre}$, and DP1$^{F/F}$/LysM$^{Cre}$ mice displayed considerably more body weight loss (Figs 4D and EV4A), higher DAI (Figs 4E and EV4A), and shorter colon (Figs 4F and EV4A) after DSS or TNBS challenge, as compared to DP1$^{F/F}$ mice. In contrast, DP1$^{F/F}$/SM22$^{Cre}$ mice did not differ from control mice in response to DSS or TNBS challenge (Figs 4D–F and EV4A). Together, these results suggest that DP1 receptor in the vascular endothelial cells, epithelial cells, and myeloid cells may mediate protective effects against DSS/TNBS-induced colitis in mice.

## Niacin suppresses vascular permeability in experimental colitis

Endothelial damage, increased colonic vascular permeability, perivascular edema, and subsequent epithelial hypoxia are essential for the development of ulcerative colitis (Deng *et al*, 2013). Deletion of DP1 receptor in vascular endothelial cells exacerbated DSS-induced colitis in mice including aggravated perivascular edema and enhanced infiltration of inflammation cells (Fig 5A). In the ear model of LPS-evoked vascular permeability, endothelial cell deficiency of DP1 significantly augmented leakage as measured by Evan's blue dye extravasation (A610: 0.27 ± 0.01 vs. 0.21 ± 0.01, $P < 0.01$, Fig 5B and C). Administration of DP1 agonist BW245C (3 mg/kg by subcutaneous injection) markedly inhibited LPS-triggered vascular permeability in DP1$^{F/F}$ mice (A610: 0.10 ± 0.01 vs. 0.21 ± 0.01, $P < 0.01$, Fig 5B and C) but not in DP1$^{F/F}$/Tie2$^{Cre}$ mice (Fig 5B and C). Consistent with this observation, more extravasated Evan's blue in colonic tissues was observed in DSS- or TNBS-challenged DP1$^{F/F}$/Tie2$^{Cre}$ mice than DP1$^{F/F}$ controls (0.26 ± 0.02 µg/mg vs. 0.11 ± 0.01 µg/mg, $P < 0.01$, Fig 5D; 0.23 ± 0.01 µg/mg vs. 0.11 ± 0.01 µg/mg, $P < 0.01$ Fig EV4B). The administration of niacin reduced Evan's blue extravasation in the inflamed intestines at both day 6 (0.10 ± 0.01 µg/mg vs. 0.13 ± 0.01 µg/mg, $P = 0.06$) and day 9 (0.14 ± 0.02 µg/mg vs. 0.26 ± 0.03 µg/mg, $P < 0.01$, Fig 5E). This reduction of vascular permeability was entirely blocked by DP1 deletion in vascular endothelial cells (Fig 5E), indicating niacin inhibits vascular permeability in intestinal tissues through activation of PGD$_2$/DP1 signal in endothelial cells.

## Niacin suppresses apoptosis of intestinal epithelial cells in experimental colitis in mice

Intestinal epithelium barrier breakdown is a hallmark of colitis. Increased apoptosis and decreased proliferation contribute to a breakdown of the epithelial barrier function in DSS-induced colitis (Araki *et al*, 2010). Indeed, DP1 deletion in intestinal epithelium (DP1$^{F/F}$/Villin$^{Cre}$) resulted in greater crypt and epithelial cell loss in mice with DSS-induced colitis as compared with DP1$^{F/F}$ mice (Fig 6A). TUNEL staining clearly showed higher frequency of apoptotic epithelial cells in DSS- or TNBS-challenged DP1$^{F/F}$/Villin$^{Cre}$ mice than in control mice (54.75 ± 4.99 cells/field vs. 24.80 ± 1.66 cells/field, $P < 0.01$, Fig 6B; 55.38 ± 3.80 cells/field vs. 16.43 ± 3.34 cells/field, $P < 0.01$, Fig EV4C), while DP1 deletion had inhibited proliferation of epithelial cells as measured by the Ki-67 immunoreactivity ($P < 0.05$, Fig 6C). Similarly, in primary cultured epithelial cells, DP1 deficiency augmented IL-13-induced epithelial apoptosis (39.65 ± 0.52% vs. 27.46 ± 0.55%, $P < 0.01$, Fig 6D). Interestingly, niacin protected colonic epithelial cells against DSS-induced apoptosis and promoted cell proliferation in WT mice, but not in DP1-deficient mice (Fig 6E–F). Thus, niacin helps maintain the intestinal epithelium barrier by activation of the DP1 receptor.

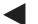

**Figure 4. DSS-induced colitis in endothelial cell, epithelial cell, macrophage, or smooth muscle cell-specific DP1-deficient mice.**

A, B   DP1 receptor expression in post-operative colon tissues from UC patients (A) and inflamed colons from DSS-challenged wild-type mice (B). The arrows indicate DP1$^+$ cells. Scale bars: 100 µm.

C   DP1 expression in endothelial cells (EC), colonic epithelial cell (EPC), peritoneal macrophages (PM), and smooth muscle cells (SMC) in tissue-specific DP1-deficient mice.

D, E   Body weight loss (D) and disease activity index (E) of DP1$^{F/F}$/Tie2$^{Cre}$ mice, DP1$^{F/F}$/Villin$^{Cre}$ mice, DP1$^{F/F}$/LysM$^{Cre}$ mice, and DP1$^{F/F}$/SM22$^{Cre}$ mice in response to DSS administration.

F   Macroscopic appearance of representative colons from DSS-challenged DP1$^{F/F}$/Tie2$^{Cre}$ mice, DP1$^{F/F}$/Villin$^{Cre}$ mice, DP1$^{F/F}$/LysM$^{Cre}$ mice, and DP1$^{F/F}$/SM22$^{Cre}$ mice (upper) and quantitation of colon length (bottom). Scale bar: 1 cm.

Data information: (C–F) Data are shown as mean ± SEM. Results are representative of two independent experiments. Statistical significance was determined using unpaired Student's *t*-tests (C) and two-way ANOVA followed by a Bonferroni *post hoc* test (D–F). (C) *$P < 0.01$; $n = 6$. (D–F) **$P < 0.01$ vs. DP1$^{F/F}$; DP1$^{F/F}$, DP1$^{F/F}$/SM22$^{Cre}$, $n = 8$; DP1$^{F/F}$/Tie2$^{Cre}$, DP1$^{F/F}$/Villin$^{Cre}$, DP1$^{F/F}$/LysM$^{Cre}$, $n = 7$.

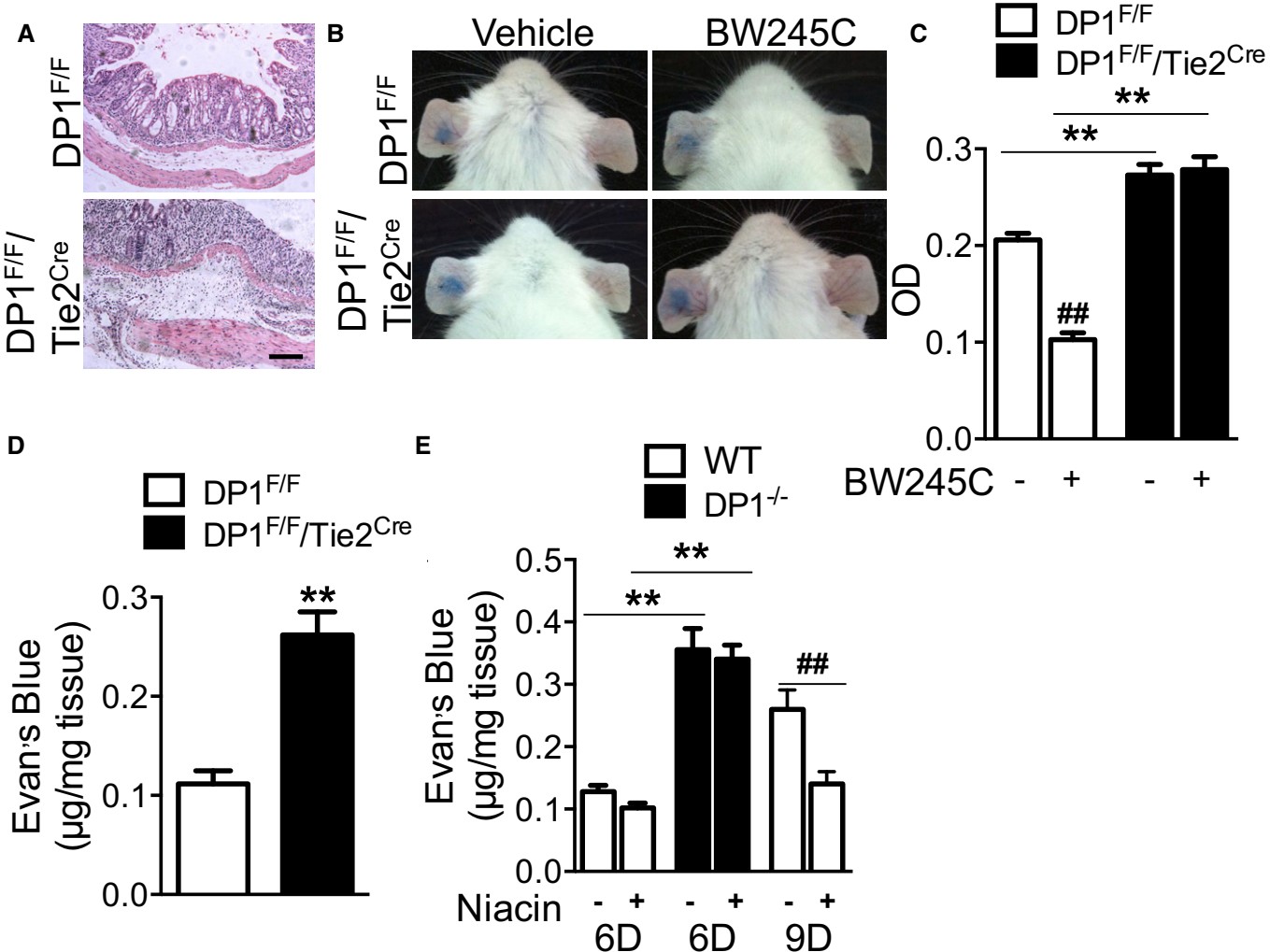

**Figure 5.  Niacin inhibits DSS-induced vascular permeability.**

A   H&E staining of distal colons from DSS-challenged DP1$^{F/F}$/Tie2$^{Cre}$ and DP1$^{F/F}$ mice. Scale bar: 100 μm.

B   Representative images of Evan's blue extravasation in ears from DP1$^{F/F}$/Tie2$^{Cre}$ and DP1$^{F/F}$ mice. Dye leakage is induced by LPS after 30 min in BW245C or vehicle pre-treated mice.

C   Quantitation of vascular permeability by measurement of dye absorbance at 610 nm in ear biopsies from DP1$^{F/F}$/Tie2$^{Cre}$ and DP1$^{F/F}$ mice. DP1$^{F/F}$-vehicle, $n = 6$; DP1$^{F/F}$-niacin, $n = 7$; DP1$^{F/F}$/Tie2$^{Cre}$-vehicle, $n = 7$; DP1$^{F/F}$/Tie2$^{Cre}$-niacin, $n = 6$.

D   Quantitative measurement of vascular permeability by dye leakage in the colonic mucosa from DSS-challenged DP1$^{F/F}$/Tie2$^{Cre}$ and DP1$^{F/F}$ mice. The mice were sacrificed at day 6. $n = 6$.

E   Effect of niacin on Evan's blue extravasation in the colonic mucosa from DP1$^{-/-}$ and WT mice. Mice were sacrificed at day 6 or day 9 as indicated. $n = 6$.

Data information: (C–E) Representative data are shown as mean ± SEM derived from two independent experiments. Statistical significance was determined using unpaired Student's $t$-tests. $^{##}P < 0.01$ vs. vehicle, $^{**}P < 0.01$ as indicated.

## Niacin depresses intestinal inflammatory reaction by promoting M2 polarization in experimental colitis in mice

Recently, we found DP1 deficiency in macrophages led to M1 polarization and delayed resolution in zymosan-induced peritonitis in mice (Kong *et al*, 2016, 2017), and deletion of the DP1 receptor in macrophages (DP1$^{F/F}$/LysM$^{Cre}$) aggravated DSS-induced colitis (Fig 4D–F). We hypothesize that activation of the DP1 receptor in macrophages (i.e., niacin intake) may reduce intestinal inflammation by directing macrophage polarization toward anti-inflammatory M2-like cells. As shown in Figs 7A and B and EV4D, disruption of

the DP1 receptor in macrophages decreased the proportion of M2-like macrophages (CD301$^+$CD68$^+$ or CD206$^+$F4/80$^+$) infiltrated in inflamed intestines in both DSS- and TNBS-challenged mice. Consistent with this finding, in intestinal macrophages separated by flow cytometry (Fig 7C), the expression of pro-inflammatory genes [tumor necrosis factor-α (TNF-α), MCP-1] is markedly induced and expression of anti-inflammatory genes (IL-4, IL-5, and IL-10) is suppressed by DP1 deficiency (Fig 7D). Interestingly, some inflammatory genes, such as IL-1β and TGF-β, displayed extremely reduced expression in intestinal macrophages (data not shown). We also examined whether niacin may delay DSS-induced colitis

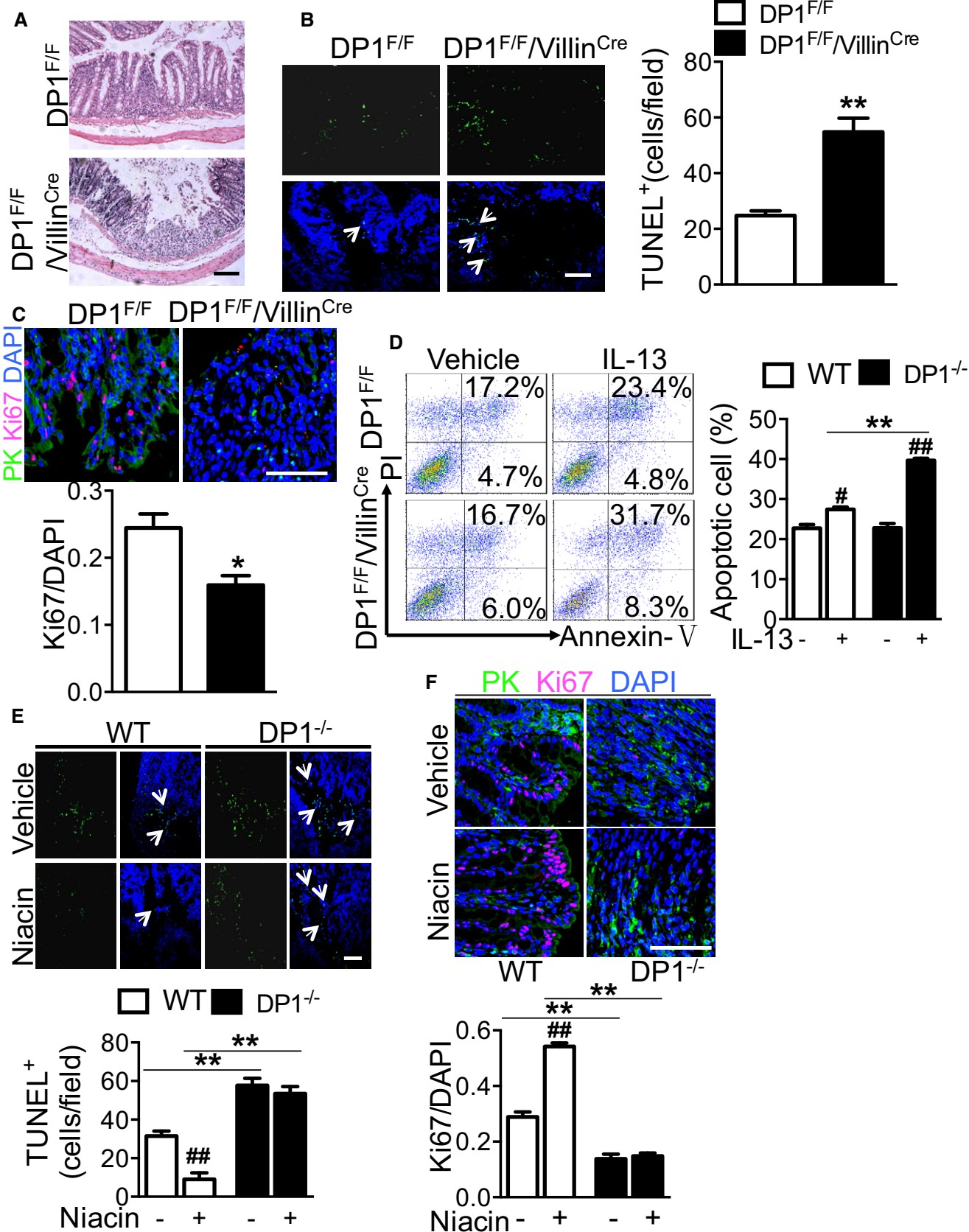

**Figure 6.**

**Figure 6.  DP1 receptor contributes to the protection of niacin against DSS-induced epithelial cell apoptosis.**

A   H&E staining on distal colons from DSS-challenged DP1$^{F/F}$/Villin$^{Cre}$ and DP1$^{F/F}$ mice. Scale bars: 100 μm.
B   TUNEL assay (left) and quantitation (right) in colonic tissues from DP1$^{F/F}$/Villin$^{Cre}$ and DP1$^{F/F}$ mice. The arrows indicate the TUNEL$^+$ cells (left). The arrows indicate the TUNEL$^+$ cells. Scale bars: 100 μm. DP1$^{F/F}$, $n = 5$; DP1$^{F/F}$/Villin$^{Cre}$, $n = 4$.
C   Representative Ki67 staining (upper) and quantitation (bottom) of Ki67 in colonic tissues from DP1$^{F/F}$/Villin$^{Cre}$ and DP1$^{F/F}$ mice. Anti-pan-keratin (PK) antibody and anti-Ki67 antibody were used. Scale bars: 100 μm. $n = 4$.
D   Representative flowcharts (left) and quantitation (right) of Annexin V-positive IL-13-treated colonic epithelial cells from DP1$^{F/F}$/Villin$^{Cre}$ and DP1$^{F/F}$ mice. $n = 3$.
E, F   Effect of niacin on DSS-induced epithelial cell apoptosis and proliferation inhibition in DP1$^{-/-}$ and WT mice. The arrows indicate the TUNEL$^+$ cells. Scale bars: 100 μm. $n = 4$.

Data information: Data are expressed as mean ± SEM. Statistical analysis was performed using unpaired Student's *t*-test. All data were verified in two independent experiments. (B, C) *$P < 0.05$, **$P < 0.01$ vs. DP1$^{F/F}$. (D) #$P < 0.05$, ##$P < 0.01$ compared with vehicle, **$P < 0.01$, as indicated. (E, F) ##$P < 0.01$ compared with vehicle, **$P < 0.01$, as indicated.

partially through modulation of macrophage polarization. Indeed, niacin facilitated intestinal macrophage polarization toward M2 status by increasing CD301$^+$CD68$^+$ cell ratio in DSS-challenged intestines (Fig 7E), downregulating pro-inflammatory genes, and provoking anti-inflammatory gene expression (Fig 7F). However, the phenotypic turnover of macrophage by niacin was not observed in myeloid DP1-deficient mice, suggesting DP1 receptor mediates the M2 polarization elicited by niacin (Fig 7G).

### PGD$_2$ infusion ameliorates DSS-induced colitis in mice

To further confirm that niacin relieves UC through releasing PGD$_2$ in mice, we directly infused PGD$_2$ to DSS-challenged mice. As shown in Fig EV5A–C, PGD$_2$ ameliorates DSS-induced colitis in mice as evidenced by decreased body weight loss, DAI, and shortening of the colon after DSS challenge. As expected, PGD$_2$ administration significantly inhibited Evan's blue extravasation in the inflamed intestines at day 9 ($0.13 \pm 0.01$ μg/mg vs. $0.25 \pm 0.02$ μg/mg, $P < 0.01$, Fig EV5D), markedly decreased DSS-induced apoptosis of epithelial cells (Fig EV5E), and increased the proportion of M2-like macrophages (Fig EV5F).

### Niacin induces clinical remission in patients with moderately active UC

We investigated whether niacin induced clinical remission in patients with moderately active UC. Twenty-six UC patients (Dubinsky *et al*, 2003; Annese *et al*, 2005), who did not respond to conventional therapies, were recruited. Demographics and baseline characteristics of patients are summarized in Appendix Table S1. Patients were assigned to receive retention enema treatment (including 300 mg niacin/100 ml) daily for 6 weeks (Fig 8A). As we expected, PGD$_2$ production as measured by urinary tetranor PGDM was elevated in patients after niacin administration (Fig 8B). PGF$_{2\alpha}$ production was also increased by niacin treatment without any influence on urinary 8-isoprostane PGF$_{2\alpha}$ (Appendix Fig S2A and B). Surprisingly, the proportion of patients with clinical response was 92.3% (24/26), and the proportion of clinical remissions was 88.5% (23/26). Compared to baseline, 23 out of 26 patients achieved mucosal healing (Fig 8C). Patients receiving niacin treatment had significant improvement in the Mayo score (Fig 8D). Each subscore, such as stool frequency, rectal bleeding, endoscopic findings, or physician's global assessment, is reduced significantly after niacin treatment (Appendix Table S2). Moreover, 24 out of 26 patients that received niacin treatment underwent overall histological improvement with normal epithelium, mucosal architecture, and lamina propia cellularity and few inflammatory cell infiltration (Fig 8E). C-reactive protein (CRP) and erythrocyte sedimentation rate (ESR) levels in plasma and platelet activities of UC patients were not markedly altered after niacin treatment (data not shown). No serious adverse effect of niacin, including flushing and urticaria, was observed (Appendix Table S3). And niacin retention enema did not influence lipid profile of patients (Appendix Table S4). Overall, enema treatment in combination with niacin is well tolerated and effective in inducing clinical remission in UC patients.

**Figure 7.  Niacin suppresses pro-inflammatory cytokine expression in macrophages in DSS-induced colitis in mice.**

A   Representative immunofluorescent staining (left) and quantitation (right) of CD301$^+$CD68$^+$ cells in colonic tissues from DSS-challenged DP1$^{F/F}$/LysM$^{Cre}$ and DP1$^{F/F}$ mice. The arrows indicate the CD301$^+$CD68$^+$ cells (left panel). Scale bar: 100 μm. $n = 4$.
B   Flow cytometry analysis of CD206$^+$F4/80$^+$ cells in colons DSS-challenged DP1$^{F/F}$/LysM$^{Cre}$ and DP1$^{F/F}$ mice. $n = 5$.
C   Flow cytometry analysis of CD11b$^+$F4/80$^+$ cells in colons from DSS-challenged mice.
D   Real-time PCR analysis of TNF-α, MCP-1, IL-4, IL-5, and IL-10 expression in colonic F4/80$^+$CD11b$^+$ cells in DP1$^{F/F}$/LysM$^{Cre}$ mice and DP1$^{F/F}$ mice. $n = 6$.
E   Effect of niacin on colonic macrophage infiltration in DSS-challenged DP1$^{-/-}$ and WT mice. The arrows indicate the CD301$^+$CD68$^+$ cells. Scale bar: 100 μm. WT-vehicle, DP1$^{-/-}$-vehicle, DP1$^{-/-}$-niacin: $n = 4$; WT-niacin, $n = 7$.
F   Effect of niacin on MCP-1, IL-5, and IL-10 expression in colonic F4/80$^+$CD11b$^+$ cells in DP1$^{-/-}$ mice and WT mice. $n = 6$.
G   Schematic illustration of the protective mechanisms of PGD$_2$/DP1 axis in UC. Niacin stimulates PGD$_2$ release in the inflamed colon tissues, which ① inhibits vascular permeability, ② suppresses DSS-induced apoptosis, and ③ downregulates pro-inflammatory cytokine secretion in macrophages through activation of DP1 receptor. IECs, intestinal epithelial cells; SMCs, smooth muscle cells; MΦ: macrophage.

Data information: All data are expressed as mean ± SEM. *P*-values were calculated using unpaired Student's *t*-test. Data are representative of at least two independent experiments. (A, B) **$P < 0.01$ vs. DP1$^{F/F}$. (D) *$P < 0.05$ vs. DP1$^{F/F}$. (E) ##$P < 0.01$ vs. vehicle, **$P < 0.01$ as indicated. (F) #$P < 0.05$ vs. vehicle, *$P < 0.05$, **$P < 0.01$ as indicated.

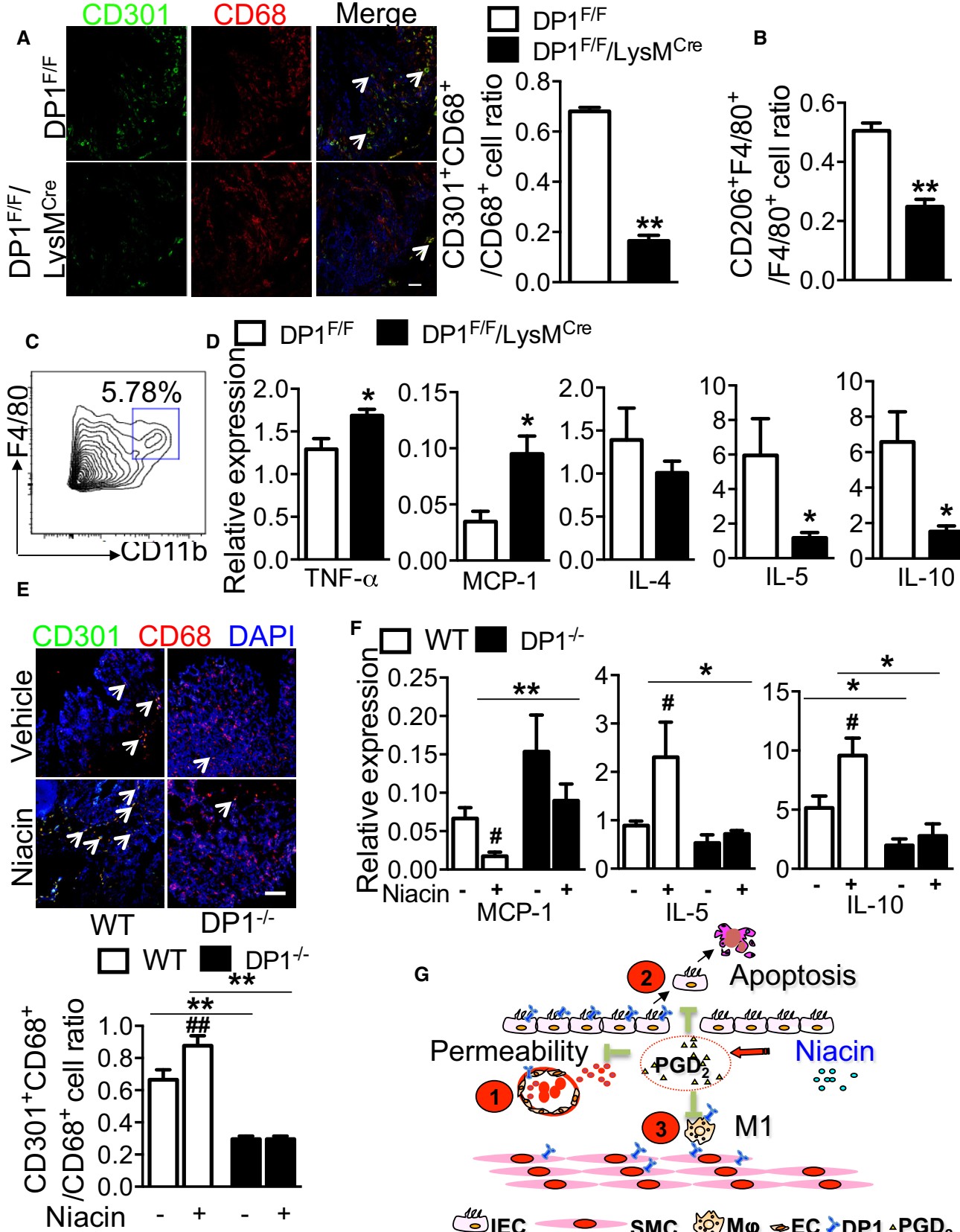

Figure 7.

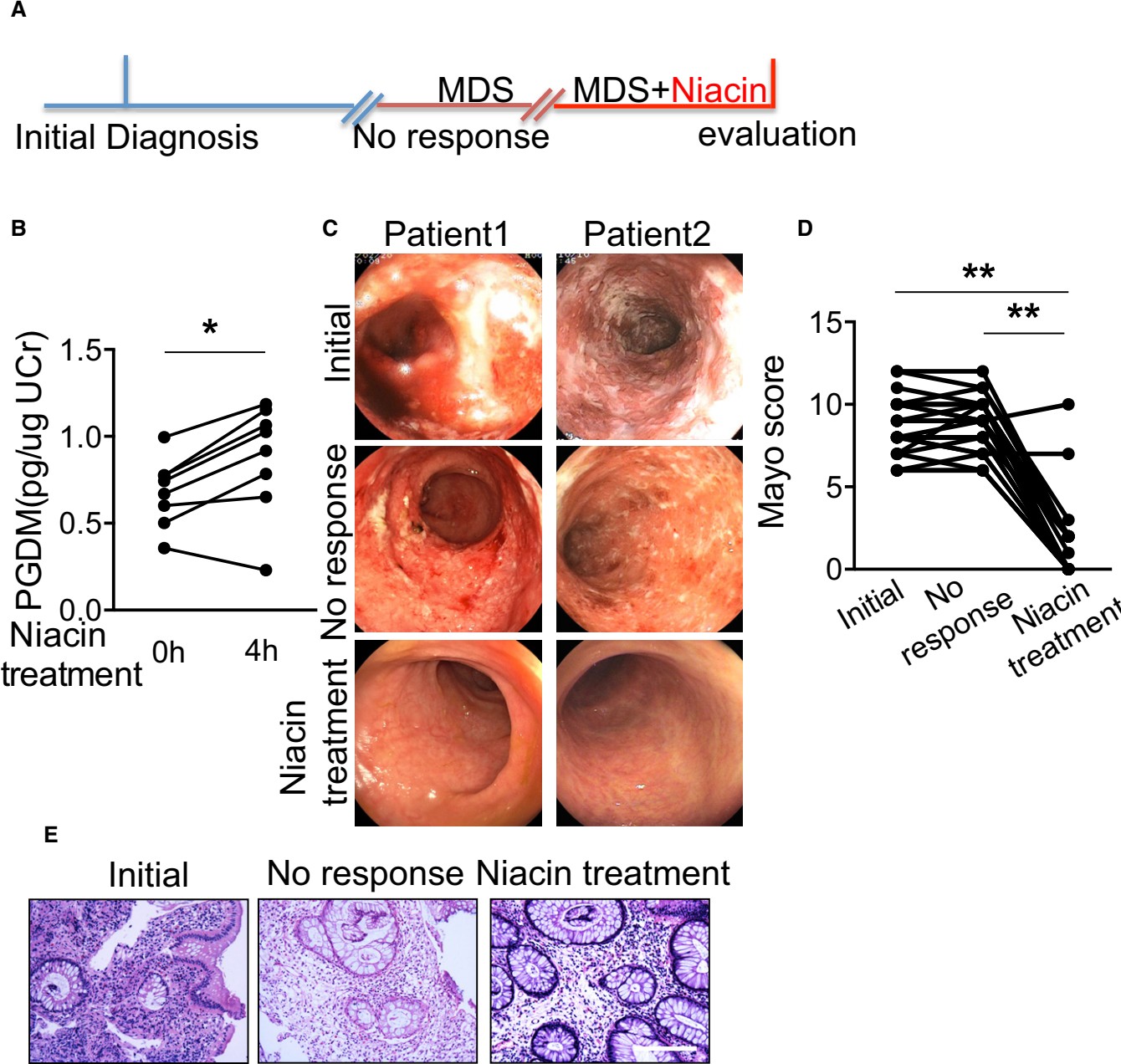

**Figure 8.  The therapeutic effect of niacin retention enema on patients with moderately active UC.**

A    Clinical study design of niacin retention enema therapy. MDS: metronidazole, dexamethasone, starch.

B    Effect of niacin retention enema on urinary tetranor PGDM production in patients.

C    Representative endoscopic images of colon mucosa from patients involved in this study. Superficial ulceration, granulocyte infiltration, and distorted/branching crypts are apparent in biopsies from patients with active disease, whereas those from the healthy subjects or those in remission appear normal.

D    Changes of Mayo score of UC patients before and after niacin retention enema therapy.

E    Representative H&E staining colon mucosa from UC patients. Scale bar: 100 μm.

Data information: *P*-values were calculated using paired Student's *t*-test. (B) **P* < 0.05 as indicated, *n* = 8. (D) ***P* < 0.01 as indicated, *n* = 26.

## Discussion

Ulcerative colitis is a chronic, relapsing inflammatory bowel disease, and the pathological changes of the colonic tissues affected include crypt branching, irregularity of size and shape of crypt, inflammatory cell infiltration in the lamina propria, and even erosion (Danese & Fiocchi, 2011). In this study, we demonstrated that niacin displayed multifaceted protective effects against DSS/TNBS-induced colitis in mice through activation of the DP1 receptor, including inhibition of vascular leakage, suppression of

    

colonic epithelium apoptosis, and reduction of pro-inflammatory cytokine secretion. These disorders interactively promote pathological progression of UC (Su *et al*, 2009; Khor *et al*, 2011). Retention enema with niacin facilitated mucosal healing in patients with moderately active UC.

Prostaglandins are synthesized from arachidonic acid through the action of phospholipases and cyclooxygenases and are involved in many inflammatory processes (Zhang *et al*, 2010; Fattahi & Mirshafiey, 2012; Dennis & Norris, 2015; Cheng *et al*, 2016). Non-steroidal anti-inflammatory drugs (NSAID), which are widely utilized as analgesic and anti-inflammatory agents for the treatment of arthritis and other inflammatory disorders, have been reported to exacerbate IBD (Felder *et al*, 2000; Bonner, 2001; Matuk *et al*, 2004). Mice lacking either COX-1 or COX-2 show increased sensitivity to DSS, and inducible COX-2-deficient mice are more susceptible than COX-1-deficient mice (Morteau *et al*, 2000). COX-2 expression in myeloid cells and endothelial cells confers protection against DSS-induced colitis (Ishikawa *et al*, 2011). Indeed, $PGD_2$ is important for induction and maintenance of UC remission in both rodents and humans (Ajuebor *et al*, 2000; Vong *et al*, 2010). Consistent with this function, blockade of $PGD_2$ downstream receptor DP1 worsens DSS-induced colitis in mice (Cheng *et al*, 2006; Sturm *et al*, 2014). We found that niacin induces remission of UC in mice via DP1. Moreover, DP1 receptor expression in vascular endothelial cells, colonic epithelium, and myeloid cells is critical to protect the colonic mucosa from injury in DSS/TNBS-induced colitis mouse models. These observations suggest that $PGD_2$ derived from COX enzymes (mostly COX-2) and perhaps $PGE_2$ (Tessner *et al*, 1998; Tanaka *et al*, 2009) are responsible for NSAID-exacerbated colitis. However, we failed to detect a significant effect of DP2 knockout on DSS-induced colitis in mice, which does not seem to match the observations obtained from pharmacological inhibition of the DP2 receptor (Sturm *et al*, 2014).

Niacin reduces plasma levels of free fatty acids and cholesterol through activation of GPR109A receptor on adipocytes while causing an unwanted facial flushing side effect due to the GPR109A-mediated $PGD_2$ release on Langerhans cells and keratinocytes (Benyo *et al*, 2006; Maciejewski-Lenoir *et al*, 2006; Hanson *et al*, 2010). Besides adipose tissue and immune cells (Feingold *et al*, 2014), GPR109A receptor is also expressed in colonic epithelium (Ganapathy *et al*, 2013). Indeed, $PGD_2$ production was induced by niacin treatment in colon tissues. The disruption of DP1 receptor augmented inflammatory response by increasing pro-inflammatory cytokine expression, enhanced apoptosis in colonic epithelium, and vascular permeability in DSS- or TNBS-challenged mice. Thus, niacin ameliorated DSS/TNBS-induced colitis by inhibition of colonic epithelium apoptosis and of inflammatory reactions in the lamina propria. In agreement with our results, niacin has been found to display anti-inflammatory properties in other inflammatory diseases such as coronary artery disease (Kuvin *et al*, 2006) and atherosclerosis (Yu & Zhao, 2007). In addition, the activation of GPR109A in myeloid cells induces IL-10 expression and subsequent differentiation of regulatory T cells (Treg cells) and IL-10-producing T cells (Singh *et al*, 2014). IL-10 secretion was dramatically reduced in DP1-deficient myeloid cells in DSS-induced colitis mice, strongly indicating that the niacin/GPR109A axis-mediated anti-inflammatory IL-10 secretion may depend on $PGD_2$/DP1 signaling in myeloid cells.

Retention enema in combination with niacin induces clinical remission and endoscopic mucosal healing of the eligible patients with moderately active UC (limited to the left half of the colon and rectum). It is an explorative and self-controlled follow-up study. However, there are several limitations of this clinical study, such as small sample size, lack of placebo control, relatively short period of medication, short follow-up period, and the fact that all the recruited patients are unresponsive to conventional treatment. Moreover, it remains to be investigated whether oral intake of niacin would have an equivalent effect on UC as the retention enema.

In summary, we demonstrated that niacin treatment ameliorates UC by boosting $PGD_2$ release in both mice and patients, indicating niacin may serve as an effective therapeutic option for UC patients, especially those with an inadequate or no response to conventional treatment.

# Materials and Methods

### Mice

Villin[Cre] mice were purchased from Model Animal Research Center of Nanjing University, and Ella[Cre], Tie2[Cre], LysM[Cre], and SM22[Cre] mice were obtained from Jackson Laboratory (Bar Harbor, Maine, USA). All the colonies including DP1 (Kong *et al*, 2016) and DP2 (Satoh *et al*, 2006) mutants were maintained on a C57BL/6 genetic background. DP1[Flox]/[Flox] (DP1[F/F]) mice were crossed with Tie2[Cre], Villin[Cre], LysM[Cre], or SM22[Cre] transgenic mice to generate cell-specific DP1-deficient mice, respectively. Animals were maintained and experiments were carried out with the approval of the Institutional Animal Care and Use Committee of the Institute for Nutritional Sciences, Chinese Academy of Sciences.

### Reagents

Dextran sulfate sodium (DSS) was purchased from MP Biomedicals (LLC, Santa Ana, California, USA). TNBS, LPS, Evan's blue, and niacin were purchased from Sigma Chemical Company (Sigma-Aldrich, St. Louis, MO, USA). $PGD_2$ and BW245C were obtained from Cayman Chemical Company (Cayman Chemical, Ann Arbor, MI, USA). Percoll solution was from Biosharp (Biosharp, Hefei, China). IL-13 was purchased from Peprotech (Peprotech, Rocky Hill, USA). TdT fluorescence *in situ* Apoptosis Detection kit was from Yeasen Biological Technology (Yeasen, Shanghai, China). Annexin V-FITC Apoptosis Detection Assay kit was obtained from Dojindo Laboratories (Dojindo, Shanghai, China).

### Induction of mouse colitis

For DSS-induced colitis, 6- to 8-week-old male mice were subjected DSS administration (molecular weight: 36,000–50,000 D) through drinking water (2%) for 6–9 days as indicated. As for TNBS-induced colitis, 6- to 8-week-old male mice were pre-sensitized with 1% TNBS at day 1 and then challenged with 2.5 % TNBS (100 μl) intrarectally at day 8 (Wirtz *et al*, 2007). Niacin was used to treat mice by gavage at a dose of 600 mg/kg/day, which is comparable to

the therapeutic dose used in dyslipidemic patients if the 10 times faster metabolism of mice as compared to humans is taken into account (van der Hoorn *et al*, 2008). Body weight was measured every day from the beginning of the DSS and TNBS administration. The body weight loss was calculated as the percentage change compared to the original body weight. The stool consistency and gross bleeding were monitored daily. The disease activity index (DAI) was calculated by the combined scores of the following parameters divided by 3: weight loss (0, normal; 1, 0–5%; 2, 5–10%; 3, 10–20%; 4, > 20%), stool consistency (0, normal; 2, loose stools; 4, watery/diarrhea), and gross bleeding in stool (0, negative; 2, positive in hemoculture; 4, macroscopic hematochezia) (Cooper *et al*, 1993).

### Primary cell culture

For preparation of peritoneal macrophages, mice were injected intraperitoneally with 1 ml of 10% thioglycollate medium (Scharlau®, Barcelona, Spain). On the fourth day, the mice were euthanized and injected intraperitoneally with 7–8 ml of ice-cold phosphate-buffered saline (PBS), and then peritoneal macrophages were collected and cultured in Roswell Park Memorial Institute (RPMI) 1,640 medium with 10% fetal bovine serum at 37°C, 5% $CO_2$ (Wang *et al*, 2014).

For preparation of colonic epithelial cells and lamina propria mononuclear cells (LPMCs), intestine biopsies were opened longitudinally and cut into 1-cm pieces. They were incubated in 5 ml pre-digestion solution Hanks' balanced salt solution (HBSS) containing 5 mM ethylenediaminetetraacetic acid (EDTA) and 1 mM dithiothreitol (DTT) for 1 h at 37°C, and the intestine pieces were subjected to 100-μm cell strainer to get single primary cells. The cells were further centrifuged with a 50% Percoll solution at 500 *g* for 20 min, and the epithelial cells were then equilibrated at the interface (Evans *et al*, 1992).

For preparation of intestinal LPMCs, intestine biopsies were cut into 1-mm pieces and incubated in a digestion solution containing 0.05 g of collagenase D (Roche, Basel, Switzerland), 0.05 g of DNase I (Sigma-Aldrich, St. Louis, MO, USA), and 0.3 g of dispase II (Roche) for about 40 min until all the pieces were fully digested. The cell pellets were obtained after centrifugation at 500 *g* for 10 min, followed by resuspension and separation in 40%/80% Percoll solution by centrifugation at 1,000 *g* for 20 min, and LPMCs could be visible as a white ring at the interface (Weigmann *et al*, 2007).

Vascular smooth muscle cells and endothelial cells were prepared from aortas and lungs as previously reported (Zhang *et al*, 2013; Lu *et al*, 2015), respectively. Vascular endothelial cells at 2–3 passages were used for experiments.

### Flow cytometry

The lamina propria mononuclear cells were stained with fluorescence-tagged primary antibodies (Brilliant Violet 421-F4/80, Biolegend, San Diego, CA, USA; FITC-anti-CD11b, MACS, Bergisch Gladbach, Germany, FITC-anti-CD206, Biolegend, San Diego, CA, USA) for 45 min at 4°C. Flow cytometry was performed using a BD LSRFortessa™ cell analyzer (BD Biosciences, San Jose, CA, USA).

### Hematoxylin and eosin and Immunofluorescence staining

For hematoxylin and eosin (H&E) staining, mouse colon tissues were fixed in 4% paraformaldehyde and embedded in paraffin and then incised a 4-μm section. The sections were stained with both hematoxylin and eosin, and then photographed using an electron microscope.

For immunofluorescence staining, colon biopsies from UC patients and mice were embedded in O.C.T. compound (Tissue Tek, Sakura, Torrance, CA, USA), and then incised 8-μm sections and processed for immunostaining. The DP1 expression was examined with polyclonal anti-DP1 primary antibody (Cayman Chemical, Ann Arbor, MI, USA). The endothelial cells were marked with anti-CD31 (1:200, BD Biosciences, San Diego, CA, USA). To detect epithelial cell and macrophage, anti-pan-keratin-FITC antibody (Cell Signaling Technology, Danvers, MA, USA) and anti-CD68 (1:200, AbD Serotec, Kidlington, UK) primary antibody were used. To label the smooth muscle cells, the anti-α-actin-FITC antibody (1:200, Sigma-Aldrich, St. Louis, MO, USA) was used. Anti-CD301 (1:100 Bio-Rad, California, USA) primary antibody was used to mark M2-like macrophage, and anti-Ki67 (1:500, Epitomics, Burlingame, CA, USA) primary antibody was used to label proliferating cell.

### Mouse genotyping

Genomic DNA from mouse tail biopsies was extracted as template. PCR products were resolved by electrophoresis in a 2% agarose gel (Biowest, Verkiu, Lithuania), and stained with ethidium bromide. The images were digitally captured with a SynGene gel image system (Tanon 2500, Shanghai, China). The primers used in this study are shown in Appendix Table S5.

### RNA preparation and RT–PCR analysis

Total RNA was isolated with TRIzol reagent (Invitrogen, San Diego, CA, USA). The reverse transcription reactions were performed by use of Reverse Transcription Reagent kits (Takara, Otsu, Shiga, Japan). Real-time PCR was conducted with SYBR Green mix (Applied Biosystems®, CA, USA). The primers used in this study are summarized in Appendix Table S5.

### Vascular permeability assay

After anesthesia, both ears of one group of mouse were injected intradermally with 3 mg/kg (in 10 μl) BW245C and the ears of the other group were injected with vehicle. After 30 min, 10 μl (1 mg/ml) LPS was injected in the left ear of all mice, whereas only PBS was injected on the right ear. 30 min later, mice received 100 μl of 1% Evan's blue in PBS through tail vein injection. All animals were euthanized after 15 min by $CO_2$ inhalation. Ear biopsies were collected with a 6-mm Acu-Punch, (Acuderm Inc., Ft. Lauderdale, FL, USA) and immersed in 1 ml of formamide overnight in a water bath at 55°C. Evan's blue dye was extracted from ear biopsies and measured by absorbance at 610 nm using a spectrophotometer.

For vascular permeability in DSS-induced UC, mice with DSS-induced colitis received intravenously 100 μl of 1% Evan's blue in PBS 15 min before being sacrificed. After autopsy, the colon

tissues were dried and weighed. Evan's blue was extracted and quantitated.

## Apoptosis analysis

For terminal deoxynucleotidyl transferase (TdT)-mediated biotin-16-dUTP nick-end labeling (TUNEL) assay, the colon tissues from mice with DSS administration were imbedded in O.C.T. compound and made into 8-μm frozen sections. The frozen sections were stained with a TUNEL fluorescence *in situ* Apoptosis Detection kit (Yeasen, Shanghai, China) according to the manufacturer's manual.

For Annexin V-FITC Apoptosis Detection, after treatment with IL-13 or vehicle, the adherent primary epithelial cells were prepared and stained with Annexin V-FITC Apoptosis Detection kit according to the manufacturer's instructions. Annexin V binding was analyzed by flow cytometry within 1 h.

## Mass spectral analysis

Urinary prostanoid metabolites, 8-isoprostane prostaglandin $F_{2\alpha}$, were extracted and quantitated as previously reported (Zhang *et al*, 2013). In brief, mouse urine was collected for 12 h in metabolic cages after niacin treatment. Samples (100 μl) were spiked with internal standards [tetranor PGDM-d6, tetranor PGEM-d6, 13,14-dihydro-15-keto-$PGF_{2\alpha}$ (PGFM)-d4, 2,3-dinor-6-keto $PGF_{1\alpha}$ (PGIM)-d4, 11-dehydro-thromboxane $B_2$ (TxM)-d4, and 8-iso-$PGF_{2\alpha}$-d4, (5 μl)] contained in acetonitrile (ACN). 200 μl methoxy-amine HCl (1 g/ml), an aqueous solution, was added. After standing for 30 min at room temperature, make capacity to 1 ml by water. The samples were applied to the cartridge conditioned with 1 ml of acetonitrile and equilibrated with 1 ml of water. After being washed with 1 ml of 5% acetonitrile in water, they were dried with vacuum for 15 min. After being subjected to elution using 1 ml of 5% acetonitrile in ethyl acetate, the samples were dissolved in 100 μl 10% ACN in water and passed through small centrifugal filters with a 0.2-μm nylon membrane prior to analysis by mass spectrometry. The urinary creatinine was used to normalize the prostaglandin metabolites, 8-isoprostane prostaglandin $F_{2\alpha}$.

Colon tissues from mice were homogenated and centrifuged and 500 μl of supernatant was collected for PG and SPM production analysis. In brief, internal standards ($PGD_2$-d4, 6-keto-$PGF_{1\alpha}$-d4, PGF2α-d4, PGE2-d4, TxB2-d4, RvD1-d5, RvE1-d5, 8-iso-PGF2α-d4 and 5(S)6(R)-LXA4-d5 (5 μl)) were added to the samples in 40 μl of citric acid (1 M) and 5 μl of 10% butylated hydroxytoluene, and then the samples were vigorously vortexed with 1 ml solvent (normal hexane:ethyl acetate, 1:1). The organic phase supernatant was collected after centrifugation (6,000 *g*, 10 min). After being subjected to elution using 1 ml of 5% acetonitrile in ethyl acetate, the samples were dissolved in 100 μl 10% ACN in water and passed through small centrifugal filters with a 0.2-μm nylon membrane prior to analysis by mass spectrometry. PG and SPM production was normalized to total protein of extracted tissues.

## Plasma lipid measurements

Plasma triglyceride (TG), total cholesterol (TC), low-density lipoprotein (LDL), and high-density lipoprotein (HDL) levels were measured using commercial kits (Jiancheng, Nanjing, China).

## Clinical study design and patients

This study was conducted to evaluate niacin as an effective alternative therapeutic agent for patients with UC unresponsive to conventional treatment. All patients were recruited from the Department of Gastroenterology of Ruijin Hospital in Shanghai Jiao Tong University School of Medicine from March 2015 to December 2016. The trial is registered with Chinese Clinical Trial Registry (ChiCTR; www.chictr.org.cn; ChiCTR-IOR-15006400). Eligible patients were 18 years old or above and had a diagnosis of moderately active left-sided UC, including the descending colon, the sigmoid colon, and rectum with a Mayo Clinic score greater than or equal to 6, a rectal bleeding subscore of 1 or higher, and endoscopic subscore of 2 or higher. Additional inclusion criteria were documentation of inadequate or no response to conventional retention enema treatment (5-aminosalicylate, metronidazole, dexamethasone, starch) and regular oral medicines in the past 1–2 years, such as 5-aminosalicylate (5-ASA) and/or corticosteroids. Briefly, eligible patients had no clinical response to following sequential therapies according to clinical practice guidelines for the medical management of ulcerative colitis within 6 months before the study (Mowat *et al*, 2011; Bressler *et al*, 2015): (i) oral 5-aminosalicylate (5-ASA, 4 g of Pentasa per day) induction therapy for 8 weeks; (ii) oral corticosteroids (30–40 mg of oral prednisolone or the equivalent per day) for 4 weeks; and (iii) oral 5-aminosalicylate (5-ASA, 4 g of Pentasa per day) with consecutive regular retention enema treatment (5 mg of dexamethasone, 0.5 g of metronidazole, and 5 g of starch in 100 ml saline per day) for 6 weeks. Patients were excluded if they had extremely severe UC, severe colonic stricture, infectious enteritis, a history of bowel surgery, major organ dysfunction, malignant neoplasm, pregnancy, hypertension, diabetes, and concomitant use of immunosuppressants such as azathioprine (AZA), mercaptopurine (MP), anti-TNF therapy. The study was reviewed and approved by the Ruijin Hospital Ethics Committee of Shanghai Jiao Tong University School of Medicine, and conducted in accordance with the Good Clinical Practice, the Belmont report, the Declaration of Helsinki, and other relevant rules and regulations. All patients provided written informed consent.

## Clinical study protocol

In the study, eligible patients received retention enema treatment with niacin (0.5 g metronidazole, 5 mg dexamethasone, 5 g starch, 300 mg niacin in 100 ml saline) daily for 6 weeks, who continued to take oral 5-aminosalicylate (4 g of Pentasa per day) throughout the study. The patients were examined at the beginning and the 6[th] week. Mayo Clinic scores were calculated, colonoscopy was performed with biopsy, and blood samples were collected for hematologic testing. In addition, the urine samples were obtained at 0 and 4 h after retention enema treatment for PG metabolite measurement. The primary endpoint was a clinical response, defined by a decrease in the Mayo Clinic score ≥ 3 points and ≥ 30% from the baseline score, with a decrease ≥ 1 point from the baseline score on the rectal bleeding subscore or a rectal bleeding subscore of 0 or 1. Clinical remission was defined as a Mayo Clinic score of 2 or lower with no subscore > 1. Mucosal healing was defined as an endoscopic subscore of 0 or 1 as assessed by a professional endoscopist (Yoshimura *et al*, 2015).

## The paper explained

### Problem

Niacin is an antidyslipidemic drug that elicits a strong flushing response through release of prostaglandin (PG) $D_2$. Ulcerative colitis (UC) is a chronic inflammatory bowel disease; however, it remains unclear whether niacin is beneficial for UC.

### Results

Niacin boosted $PGD_2$ release *in vivo* and improved both DSS- and TNBS-induced colitis in mice via the D prostanoid receptor 1 (DP1). DP1 expression varied between vascular wall, colonic epithelium, and infiltrated macrophages in the inflamed colons of both humans and mice. DP1 receptor deficiency in vascular endothelial cells, colonic epithelium, and myeloid cells intensified the DSS- or TNBS-induced colitis in mice through increasing vascular permeability, promoting apoptosis of epithelial cells, and stimulating pro-inflammatory cytokine secretion from macrophages, respectively. Niacin treatment improved vascular permeability, reduced apoptosis of epithelial cells, and suppressed pro-inflammatory cytokine expression from macrophages. Moreover, treatment with niacin-containing retention enema effectively promoted UC clinical remission and mucosal healing in patients with moderately active disease.

### Impact

Niacin displays multiple beneficial effects on colitis in mice and humans by activation of the $PGD_2$/DP1 axis. These results suggest niacin may become an effective therapeutic option for UC patients.

## Statistical analysis

All data are expressed as the mean ± standard error of the mean (SEM). Data were analyzed using GraphPad Prism software, version 5.0 (GraphPad Software, San Diego, CA, USA). The two-tailed unpaired Student's *t*-test was performed to compare the two datasets. Multiple comparisons were tested with two-way ANOVA followed by Bonferroni's post-test. A *P*-value of less than 0.05 was considered statistically significant. For the clinical trial, paired Student's *t*-test was used to compare the values before and after niacin treatment. The exact *P*-values in each figure are listed in Appendix Table S6.

**Expanded View** for this article is available online.

## Acknowledgements

This work was supported by the National Natural Science Foundation of China (81525004, 91439204, 81272263, 81672719) and the Science and Technology Commission of Shanghai Municipality (15140902000, 14JC1407400). Ying Yu is a fellow at the Jiangsu Collaborative Innovation Center for Cardiovascular Disease Translational Medicine.

## Author contributions

JL, LWa, and Ying Yu designed research; JL, DK, QW, WW, YT, TB, LG, LWe, QZha, Yu Yu, YQ, SZ, GL, QL, SW, YZ, YW, QZhu, DJ and WY performed research and analyzed data; YJ, HY, MN, ML, and RMB contributed experimental reagents; JL, LWa, and Ying Yu wrote the manuscript.

## Conflict of interest

The authors declare that they have no conflict of interest.

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
