## [Review Process File · EMBO Molecular Medicine]

Niacin ameliorates ulcerative colitis via prostaglandin D2-mediated D prostanoid receptor 1 activation

Juanjuan Li, Deping Kong, Qi Wang, Wei Wu, Yanping Tang, Tingting Bai, Liang Guo, Lumin Wei, Qianqian Zhang, Yu Yu, Yuting Qian, Shengkai Zuo, Guizhu Liu, Qian Liu, Sheng Wu, Yi Zang, Qian Zhu, Daile Jia, Yuanyang Wang, Weiyao Yao, Yong Ji, Huiyong Yin, Masataka Nakamura, Michael Lazarus, Richard M. Breyer, Lifu Wang, Ying Yu

*Corresponding authors: Lifu Wang, Shanghai Jiao Tong University School of Medicine
Ying Yu, Shanghai Institutes for Biological Sciences*

Review timeline:

Submission date:	28 August 2016
Editorial Decision:	19 October 2017
Revision received:	14 January 2017
Editorial Decision:	07 February 2017
Revision received:	15 February 2017
Accepted:	16 February 2017

Transaction Report:

Editor: Céline Carret

1st Editorial Decision

19 October 2017

Thank you for the submission of your manuscript to EMBO Molecular Medicine. We have now heard back from the three referees whom we asked to evaluate your manuscript. Although the referees find the study to be of potential interest, they also raise a number of concerns that need to be addressed in the next final version of your article.

You will see from comments pasted below, that all referees agree that the study is of interest and has clear translational potentials. However, mechanistic details are missing and in several instances more details and clarifications are needed. Further experiments regarding Niacin treatment in animals but also in patients would improve not only conclusiveness but also the clinical impact. Referee 2 further recommended that as you have "collected urine samples before and after treatment with niacin, assessing (in addition to PGDM) systemic markers of prostanoid biosynthesis such as PGIM, PGEM and 11-dehydro-TXB2 (TXM) they are enzymatic metabolites of PGI2, PGE2 and TXB2, respectively, would be desirable. These measurements would help clarify the mechanism of action of the drug. In particular, to have information on the cellular target of niacin. In addition, measuring the urinary levels of 8-iso-PGF2a, as a marker of oxidative stress would be informative regarding the action of niacin through an antioxidant mechanism. Finally, it is quite important to know if patients were dyslipidemic and whether niacin affected plasma lipid profile. "

Given the balance of these evaluations, we feel that we can consider a revision of your manuscript if

you can address the issues that have been raised within the space and time constraints outlined below. Please note that it is EMBO Molecular Medicine policy to allow only a single round of revision and that, as acceptance or rejection of the manuscript will depend on another round of review, your responses should be as complete as possible.

Revised manuscripts should be submitted within three months of a request for revision; they will otherwise be treated as new submissions, except under exceptional circumstances in which a short extension is obtained from the editor.

Please read below for important recommendations upon submission of the revised article.

I look forward to seeing a revised form of your manuscript as soon as possible.

***** Reviewer's comments *****

Referee #1 (Comments on Novelty/Model System):

Methods of analysis are somewhat incomplete with respect to lipid mediator analysis, which is undefined and can be clarified. Mechanisms of IBD with regard to PGD2 are founded in citations provided, but are expanded on herein. Human data are provided for translational impact but lacks a significant sample size, which could be expanded.

Referee #1 (Remarks):

Comments

The manuscript by Li et al. provides a basis for the use of niacin as a treatment for IBD. Of interest, the authors identified a link between niacin and the prostaglandin D2 receptor, DP1, which provided protection against IBD via multiple cell specific mechanisms. Overall, the evidence is clearly presented and is of wide interest to the readers of EMBO Molecular Medicine. Several major and minor concerns/suggestions are detailed below.

Major concerns/suggestions:

- 1) While the authors demonstrate that PGD2 and its metabolites (PGDM) are increased after niacin administration, there is no mention of the method by which this was determined. Was PGD2 and PGDM measured by ELISA or LC-MS/MS? This should be clarified in the results, SI Figure 1 and methods. Also, which metabolites of PGD2 were measured? There are numerous PGD2 non-enzymatic breakdown products and metabolites, such as PGJ2, 15-deoxy-PGD2, as well as enzymatic metabolites. The identity of the PGDM should be identified in the results and SI Figure 1 legend.
- 2) The classification of PGD2 as a "pro-resolution mediator" should be removed since PGD2 and PGE2 are both widely appreciated to be pro-inflammatory mediators. These PGs and others have been shown to have pleiotropic effects that are both pro-inflammatory and sometimes anti-inflammatory. Of importance here, PGE2 and PGD2 have been shown to indirectly promote resolution through induction of 15-lipoxygenase, which increases the production of specialized pro-resolving mediators (SPMs) including lipoxin A4. (Levy, et al. Nature Immunology, 2001, PMID: 11429545).
- 3) Based on the point above, the authors should strongly consider determining whether SPMs, such as lipoxin A4, resolvin D1, or other SPMs correlate with niacin effects along with PGD2 which may provide a more complete mechanism.

Minor concerns/suggestions:

- 1) The authors refer to dextran sulfate sodium (DSS) administration as "DSS treated" at one point, which creates confusion as to whether DSS is used as an inflammatory stimulus or a therapy. "DSS treated" should be replaced with "DSS administration" or some variation for clarity wherever used

throughout the manuscript.

2) Since the increase in PGD2 and its metabolites provides the mechanism by which niacin evokes therapeutic effects through DP1, Supplementary Figure 1 should be moved from the supplementary section to Figure 1.

3) Discussion, line 10: When referencing the role of prostaglandins, cyclooxygenases and phospholipases in inflammation, the authors should include more encompassing and recent references, e.g. Dennis and Norris, *Nature Reviews of Immunology*, 2015, PMID: 26139350.

4) A dose curve should be performed with niacin to demonstrate specificity, for example, on macrophage polarization and/or PGD2 increase in IBD. PGD2 should also be administered in IBD and/or permeability assays and/or macrophage polarization assays to prove PGD2-DP1 actions directly.

5) Figure 4 legend. Please specify the day(s) on which Evan's blue extravasation was measured. This is clarified in the results for experiments detailed in Supplementary Figure 4 but not in the main Figure 4.

Referee #2 (Remarks):

In the present manuscript, Li et al. studied whether Niacin which is known to induce the release of prostaglandin (PG)D2, a pro-resolution mediator, is beneficial for the treatment of inflammatory bowel disease (IBD). They found that niacin administration enhanced PGD2 production in colon tissues in dextran sulfate sodium (DSS)-treated mice, and protected mice against DSS or 2,4,6-trinitrobenzene sulfonic acid (TNBS)-induced colitis in a DP1-dependent manner. By performing experiments with mice carrying the specific deletion of the DP1 receptor in vascular endothelial cells, colonic epithelium and myeloid cells they were able to show that the receptor acts by mitigating DSS/TNBS-induced colitis. DP1 activation plays a role by restraining several responses associated with the disease, i.e., increasing vascular permeability, apoptosis of epithelial cells and enhanced pro-inflammatory cytokine secretion by macrophages. They provide evidence that Niacin treatment affected these response associated with colitis through a mechanism involving PGD2-DP1 dependent signaling pathway. They performed a small clinical study in UC patients with moderately active disease treated with niacin-containing retention enema, and it was found that the drug promoted remission and mucosal healing.

This is an interesting study, and the results are potentially clinically relevant. However, I raised some points to be addressed by the authors.

Major points

-Experimental colitis has been shown to be associated with altered plasma lipids, including an increase in plasma cholesterol. On the other side, niacin has a broad range of effects on serum lipids and lipoproteins, including lowering total cholesterol, low-density-lipoprotein (LDL) cholesterol, and triglycerides. Thus, it is important to report data on plasma lipids in experimental colitis before and after niacin treatment.

-In experimental colitis, the authors used niacin at a dose 600 mg/kg by gavage, once a day; is this dose related to that used in the in vivo study? Moreover, it is important to report whether this is comparable to the therapeutic doses used in dyslipidemic patients.

-Enhanced oxidative stress has been reported in experimental colitis, and niacin could act, at least in part, through an anti-oxidant mechanism. Thus, it is of interest to assess mucosal and urinary levels of F2-isoprostanes, as a marker of oxidative stress, before and after niacin treatment of mice.

-They should evaluate whether niacin enhanced urinary levels of PGD-M.

-Previous studies have revealed that niacin evoked platelet COX-1-derived PGD2 biosynthesis, as

assessed by measuring urinary levels of PGD-M. Platelets may be activated in inflammatory colitis. Thus, it is important to verify whether some of the beneficial effects of niacin were dependent on the inhibition of platelet activation.

-They should report in a Table the data of body mass index, total cholesterol, HDL-C and LDL-C of patients at baseline. The possible presence of other cardiovascular risk factors. Importantly, they should report if niacin altered the plasma lipid profile.

-They should report sample-size calculation of the clinical study.

- Similarly to the comments related to the experimental model of colitis, it would be important to assess the urinary levels F2-isoprostanes before and after niacin treatment.

They administered 300 mg of niacin daily for six weeks. As for the studies in mice, the authors should comment if this dose is used to treat dyslipidemia.

Referee #3 (Remarks):

In this study Li et al. investigate the role PGD2 in the protective effects of niacin treatment in two colitis models in mice and in ulcerative colitis in humans. It was found that DP1, but not DP2 receptors, in endothelium, colonic epithelial cells, and myeloid cells were responsible for the beneficial effects of niacin treatment. Moreover, niacin was effective when given to a small cohort of ulcerative colitis patients as retention enema in increasing urine PGD2 metabolites and in reducing clinical symptoms. The study is well designed and technically sound, although some aspects of tissue-specific deletion have not been properly addressed. In addition, while DP1 was found as one downstream effector of niacin, the mechanism of action has not been addressed. Specific concerns are:

1. The main question remains how Niacin increases PGD2 production. Because this is not addressed in this study, the mechanism of action remains elusive. The authors investigated how PGD2 promotes intestinal homeostasis by activating the DP1 receptor. However, the mechanisms by which niacin induces PGD2 production have not been explored. Questions to be asked in this context is whether niacin-induced changes in the lipid metabolism changes PLC activities and arachidonic acid metabolism, COX-1 and/or COX-2 activities, or whether there is a specific effect on PGD synthases. What happens to other prostanoids? Addressing some of these questions will give more mechanistic insights on the mechanism of niacin actions. In this regard, the title of the manuscript is somewhat misleading as it implies a direct action of niacin on the DP1 receptor.

2. The niacin-mediated increase in PGD2 should be presented as main data.

3. The experiments on endothelial-specific DP1 KO using the Tie2-Cre deleter mouse are not convincing. Tie2-Cre will delete in most hematopoietic cells including myeloid cells. The fact that the authors did not observe deletion in peritoneal macrophages might be due to their non-hematogenic origin from yolk sac precursors (Cassado et al, Front Immunol. 2015). In contrast, intestinal macrophages are continuously replenished from circulating monocytes, especially under inflammatory conditions, and might therefore be DP1 deleted. It is therefore necessary to confirm the status of the DP1 genetic locus in intestinal macrophages isolated from Tie2-Cre mice with colitis by flow sorting.

4. The cause-effect relationships of vascular permeability, epithelial apoptosis, and inflammatory signaling in myeloid cells has not been addressed. How do these alterations contribute to the pathology in a DP1-dependent manner? For example, is vascular permeability and epithelial apoptosis also altered in LysM-Cre mice because they have less inflammatory mediators and, vice versa, how is the vascular permeability and myeloid inflammatory response altered in Villin-Cre mice?

5. Figure 2F: The time window is too short to monitor an effect of niacin on survival in WT mice.

6. Figure 6A: Quantification for these surface markers would be more reliable if done by flow cytometry.

7. There is no placebo group included in the clinical study. This is a major omission.

1st Revision - authors' response

14 January 2017

Referee #1 (Comments on Novelty/Model System):

Methods of analysis are somewhat incomplete with respect to lipid mediator analysis, which is undefined and can be clarified. Mechanisms of IBD with regard to PGD₂ are founded in citation provided, but are expanded on herein. Human data are provided for translational impact but lacks a significant sample size which could be expanded.

Thank you very much for your critiques for further improvement of our manuscript. As suggested, we expanded the Methods to clarify lipid mediator analysis (see **Responses to Major concerns 1**) and updated the sample size (from 18 to 26; see **new Figure 8D, Table EV1-EV4**).

Indeed, PGD₂ production is markedly elevated in inflamed colon tissues from both ulcerative colitis (UC) patients and experimental colitis murine models [*Proc Natl Acad Sci U S A* 2010;107:12023-7; *Am J Physiol Gastrointest Liver Physiol* 2000;279:G238-44.], which is associated with long-term remission in humans ENREF 22 [*Proc Natl Acad Sci U S A* 2010;107:12023-7], and blockade of PGD₂ receptor DP1 worsens DSS-induced colitis in mice [*Proc Natl Acad Sci U S A* 2006;103:6682-7]. In this study, we found, niacin-an old antidiabetic drug, ameliorates UC in mice and humans through releasing PGD₂ and acting on DP1 receptor particularly in vascular endothelial cells, colonic epithelium and myeloid cells in inflamed colon tissues.

Referee #1 (Remarks):

The manuscript by Li et al. provides a basis for the use of niacin as a treatment for IBD. Of interest, the authors identified a link between niacin and the prostaglandin D2 receptor, DP1, which provided protection against IBD via multiple cell specific mechanisms. Overall, the evidence is clearly presented and is of wide interest to the readers of EMBO Molecular Medicine. Several major and minor concerns/suggestions are detailed below.

Major concerns/suggestions:

1) While the authors demonstrate that PGD₂ and its metabolites (PGDM) are increased after niacin administration, there is no mention of the method by which this was determined. Was PGD₂ and PGDM measured by ELISA or LCMS/MS? This should be clarified in the results, SI Figure 1 and methods. Also, which metabolites of PGD₂ were measured? There are numerous PGD₂ non-enzymatic breakdown products and metabolites, such as PGJ₂, 15-deoxy-PGD₂, as well as enzymatic metabolites. The identity of the PGDM should be identified in the results and SI Figure 1 legend.

I apologize for the confusion. All PG products and metabolites in our study were routinely measured by mass spectrometry. And PGD₂ metabolite (PGDM) in urine is 11,15-Dioxo-9-hydroxy-2,3,4,5-tetranorprostan-1,20-dioic acid (tetranor PGDM). As suggested, we expanded the Methods (**Expanded View Materials Page 4, Paragraph 3**), mentioned in both Results (**Page 6, Line 5**), Figure legend (**Page 28, Line 8 and Expanded View Materials, Page 7, Lines, 7-8**).

2) The classification of PGD₂ as a “pro-resolution mediator” should be removed. Since PGD₂ and PGE₂ are both widely appreciated to be pro-inflammatory mediators. These PGs and others have been shown to have pleiotropic effects that are both pro-inflammatory and sometimes anti-inflammatory mediators. Of importance here, PGE₂ and PGD₂ have been shown to indirectly promote resolution through induction of 15-lipoxygenase, which increases the production of specialized pro-resolving mediators (SPMs) including lipoxinA₄. (Levy, et al. *Nature Immunology*, 2001, PMID: 11429545).

Thanks, we removed. please see **Page 2, Line 3 and Page 13, Line 22**.

3) Based on the point above, the authors should strongly consider determining whether SPMs, such as lipoxin A₄, resolvin D1, or other SPMs correlate with niacin effects along with PGD₂ which may provide a more complete mechanism.

As suggested, we examined Resolvin (Rv) D1, E1 and lipoxin (LX) A4 products in colon tissues from niacin-treated mice after DSS challenge using mass spectrometry approach. Niacin had no significant influence on RvE1 and LXA4 production in colon tissues in DSS-challenged mice. RvD1 is undetectable in colon tissues. We expanded the results, please see **Figure EV1C** and **Page 6, Lines, 16-17**.

Minor concerns/suggestions:

1) The authors refer to dextran sulfate sodium (DSS) administration as "DSS treated" at one point, which creates confusion as to whether DSS is used as an inflammatory stimulus or a therapy. "DSS treated" should be replaced with "DSS administration" or some variation for clarity wherever used throughout the manuscript.

Thanks. We corrected throughout the manuscript.

2) Since the increase in PGD2 and its metabolites provides the mechanism by which niacinevo kes therapeutic effects through DP1, Supplementary Figure 1 should be moved from the supplementary section to Figure 1.

As suggested, we moved Supplementary Figure 1 to Figure 1, please see **new Figure 1**

3) Discussion, line 10: When referencing the role of prostaglandins, cyclooxygenases and phospholipases in inflammation, the authors should include more encompassing and recent references, e.g. Dennis and Norris, Nature Reviews of Immunology, 2015, PMID: 26139350.

As suggested, we added, please see **Page 13, Line 14**.

4) A dose curve should be performed with niacin to demonstrate specificity, for example, on m acrophage polarization and/or PGD2 increase in IBD. PGD2 should also be administered in IB D and/or permeability assays and/or macrophage polarization assays to prove PGD2-DP1 actions directly.

These are great questions. As suggested, we amended, i) dose-dependent effect of niacin on PGD₂ production in colon tissues and PGD₂ metabolites in urine in mice (**Figure 1A-B**); ii) effect of PGD₂ infusion on UC in DSS-challenged mice, including macrophage polarization, vascular permeability and epithelium apoptosis (**Figure EV6A-F**). We expanded the results, please **Page 6, Line 9** and **Page 11, Paragraph 1**.

Moreover, we recently found niacin and PGD₂ promotes M2 macrophage polarization through suppression of /JAK2/STAT1 pathway at dose-dependent manner [*J Exp Med.* 2016;213:2209-26; *J Pharmacol Exp Ther.* 2017. pii: jpet.116.238261]. We mentioned in the Results and cited these Refs (see **Page 10, Lines, 8-9**)

5) Figure 4 legend. Please specify the day(s) on which Evan's blue extravasation was measured. This is clarified in the results for experiments detailed in Supplementary Figure 4 but not in the main Figure 4.

We added, please see **new Figure 5 legend, Page 30, Lines, 19 and 21**.

Referee #2 (Remarks):

In the present manuscript, Li et al. studied whether Niacin which is known to induce the release of prostaglandin (PG)D₂, a pro-resolution mediator, is beneficial for the treatment of inflammatory bowel disease (IBD). They found that niacin administration enhanced PGD₂ production in colon tissues in dextran sulfate sodium (DSS)-treated mice, and protected mice against DSS or 2,4,6,- trinitrobenzene sulfonic acid (TNBS)-induced colitis in D prostanoid receptor 1 (DP1) - dependent manner. By performing experiments with mice carrying the specific deletion of the DP1 receptor in vascular endothelial cells, colonic epithelium and myeloid cells, they were able to show that the receptor act by mitigating DSS/TNBS induced colitis. DP1 activation plays a role by restraining several responses associated with the disease, i.e., increasing vascular permeability, apoptosis of epithelial cells and enhanced pro-inflammatory cytokine secretion by macrophages. They provide evidence that Niacin treatment affected these response associated with colitis through a mechanism involving PGD₂-DP1 dependent signaling pathway. They performed a small clinical study in UC patients with moderately active disease treated with niacin-containing retention enema, and it was found that the drug promoted remission and mucosal healing. This is an interesting study, However, I raised some points to be addressed by the authors.

Major points

1. Experimental colitis has been shown to be associated with altered plasma lipids, including an increase in plasma cholesterol. On the other side, niacin has a broad range of effects on serum lipids and lipoproteins, including lowering total cholesterol, low-density-lipoprotein (LDL) cholesterol, and triglycerides. Thus, it is important to report data on plasma lipids in experimental colitis before and after niacin treatment.

As suggested, we measured plasma lipid levels in experimental colitis before and after niacin treatment. Please see **Page 7, Lines, 17-20**, and **Figure EV4B**.

2. In experimental colitis, the authors used niacin at a dose 600 mg/kg by gavage, once a day; is this dose related to that used in the in vivo study? Moreover, it is important to report whether this is comparable to the therapeutic doses used in dyslipidemic patients.

We apologize for the confusion. A dose 600 mg/kg/d of niacin were routinely used for experimental colitis in mice as previously described elsewhere in animals [*Arterioscler Thromb Vase Biol.* 2008;28:2016-2022; *Pharmacol Biochem Behav.* 2012;101:493-8]. For patients with UC, 300 mg niacin (in 100 ml saline) was used for retention enema treatment daily.

In mice, niacin lowers plasma triglycerides and total cholesterol at dose-dependent fashion (118 mg/kg/d to 1180 mg/kg/d). These doses for mice are safe and correspond well to the doses used in humans (3-6 g/d for adults), if the 10-time faster metabolism of mice as compared to humans is taken into account [*Arterioscler Thromb Vase Biol.* 2008;28:2016-2022]. We mentioned in the Method (**Page 16, Lines, 18-20**).

3. Enhanced oxidative stress has been reported in experimental colitis, and niacin could act, at least in part, through an anti-oxidant mechanism. Thus, it is of interest to assess mucosal and urinary levels of F₂-isoprostanes, as a marker of oxidative stress, before and after niacin treatment of mice.

Thanks for the insightful comments. As requested, we examined the F₂-isoprostanes in both colon tissues and urine, and did not observe notable changes after niacin treatment in mice. We expanded the results, please see **Page 7, Lines, 17-19**, and **Figure EV4A**

4. They should evaluate whether niacin enhanced urinary levels of PGD-M.

We did, please see details in **Figure 1B**.

5. Previous studies have revealed that niacin evoked platelet COX-1-derived PGD₂ biosynthesis, as assessed by measuring urinary levels of PGD-M. Platelets may be activated in inflammatory colitis. Thus, it is important to verify whether some of the beneficial effects of niacin were dependent on the inhibition of platelet activation.

This is a great question. It is true niacin evokes platelet COX-1-derived PGD₂ biosynthesis in human and in mice, and niacin suppresses human platelet activity by DP1 receptor [*J Clin Invest.* 2012;122:1459-68]. Although vascular expression of DP1 is conserved between human and mice, platelet DP1 receptor is not present in mice [*Circulation.* 2001;104:1176-1180; *J Clin Invest.* 2012;122:1459-68]. Given platelets may be activated in inflammatory colitis [*Gastroenterology.* 2007;132:955-65. *Gut.* 2003;52:1435-41.], we examined platelet activity from UC patients before and after niacin retention enema, and did not detect notable impact of niacin treatment (see below **Reviewer Figure 1**, [UNPUBLISHED DATA REMOVED PER AUTHORS REQUEST]). We expanded the results, see **Page 11, Lines, 27-29**.

6. They should report in a Table the data of body mass index, total cholesterol, HDL-C and LDL-C of patients at baseline. The possible presence of other cardiovascular risk factors. Importantly, they should report if niacin altered the plasma lipid profile.

As suggested, i) we added the data of body mass index in **Table EV1**; summarized blood routine tests (such as cholesterol, HDL-C and LDL-C and white blood cells, neutrophils, albumin and hemoglobin levels) of patients before and after niacin treatment in **Table EV4**. ii) patients with other cardiovascular risk factors (such as hypertension, diabetes) were excluded. (see **Page 17, Lines 24-25**). Interestingly, niacin retention enema treatment did not alter lipid profile of patients (**Table EV4**), probably due to local delivery and insufficient drug concentration.

7. They should report sample-size calculation of the clinical study.

Thank you for your comment. This clinical trial using niacin retention enema is a self-controlled follow-up study with relative small sample size. It is an explorative pilot study, sample size was

determined based on previously self-controlled trials on patients with UC [*Am J Gastroenterol.* 2003;98:1058-63; *Am J Gastroenterol.* 2005;100:1370-5.]. We updated the sample size (Table EV1-4) and discussed the limitations, please Response to Reviewer 3 Question 7 and Page 15, last Paragraph.

8. Similarly to the comments related to the experimental model of colitis, it would be important to assess the urinary levels F2-isoprostanes before and after niacin treatment. We did, see Page 11, Lines, 17-18 and Figure EV7B.

9. They administered 300 mg of niacin daily for six weeks. As for the studies in mice, the authors should comment if this dose is used to treat dyslipidemia.

We apologize for the confusion. We used 600mg niacin/kg/d for animal study (please see Responses to Reviewer 2 Question 2). Patients were received retention enema treatment for 6 weeks (300 mg niacin in 100 ml enema). We modified the sentence, please see Page 11, Line 15.

Referee #3 (Remarks):

In this study Li et al. investigate the role PGD2 in the protective effects of niacin treatment in two colitis models in mice and in ulcerative colitis in humans. It was found that DP1, but not DP2 receptors, in endothelium, colonic epithelial cells, and myeloid cells were responsible for the beneficial effects of niacin treatment. Moreover, niacin was effective when given to a small cohort of ulcerative colitis patients as retention enema in increasing urine PGD2 metabolites and in reducing clinical symptoms. The study is well designed and technically sound, although some aspects of tissue-specific deletion have not been properly addressed. In addition, while DP1 was found as one downstream effector of niacin, the mechanism of action has not been addressed. Specific concerns are:

1. The main question remains how Niacin increases PGD2 production. Because this is not addressed in this study, the mechanism of action remains elusive. The authors investigated how PGD2 promotes intestinal homeostasis by activating the DP1 receptor. However, the mechanisms by which niacin induces PGD2 production have not been explored. Questions to be asked in this context is whether niacin-induced changes in the lipid metabolism changes PLC activities and arachidonic acid metabolism, COX-1 and/or COX-2 activities, or whether there is a specific effect on PGD synthases. What happens to other prostanoids? Addressing some of these questions will give more mechanistic insights on the mechanism of niacin actions. In this regard, the title of the manuscript is somewhat misleading as it implies a direct action of niacin on the DP1 receptor.

Thanks for the insightful comments. Niacin induces PGD₂ secretion through activation of cytosolic phospholipase A2 (cPLA₂) [*J Clin Invest.* 2009; 119(5): 1312–1321] and both COX-1 and COX-2 in immune cells [*J Clin Invest.* 2005;115:3634-40] and keratinocytes [*J Clin Invest.* 2010;120:2910-9], which are all mediated by its GPR109A [*J Clin Invest.* 2005;115:3634-40]; *J Clin Invest.* 2009; 119: 1312–1321]. As such, we examined the expression of cPLA₂, COX-1, COX-2, hematopoietic PGD synthase (hPGDS), lipocalin-type PGD synthase (lPGDS) in peritoneal macrophages followed niacin treatment. Indeed, expression of cPLA₂, COX-2 and hPGDS were significantly increased in macrophages after niacin treatment, while lPGDS was undetectable. We expanded the results, see Page 6, Lines, 13-15, and Figure 1C-E.

We also amended the changes of other PGs in colon tissues and urinary PG metabolites in mice after niacin treatment. Please see Page 6, Lines, 10-12, and Figure EV1A-1B.

As suggested, we modified the title of the manuscript.

2. The niacin-mediated increase in PGD2 should be presented as main data.

Thanks. We moved Supplementary Figure 1 to new Figure 1.

3. The experiments on endothelial-specific DP1 KO using the Tie2-Cre deleter mouse are not convincing. Tie2-Cre will delete in most hematopoietic cells including myeloid cells. The fact that the authors did not observe deletion in peritoneal macrophages might be due to their non-hematogenic origin from yolk sac precursors (Cassado et al, *Front Immunol.* 2015). In contrast, intestinal macrophages are continuously replenished from circulating monocytes, especially under inflammatory conditions, and might therefore be DP1 deleted. It is necessary to confirm the status of the DP1 genetic locus in intestinal macrophages isolated from Tie2-Cre mice with colitis by flow sorting.

Thanks for the critique. We agree, Tie2 kinase is also expressed in hematopoietic progenitors, which leads to Cre recombinase leakage in hematopoietic cells (including myeloid cells) in Tie2^{Cre} tool mice [*Proc Natl Acad Sci U S A.* 2009;106:10266-71; *J Exp Med.* 2001;193:741-754]. As suggested, we analyzed DP1 expression in CD11b⁺F4/80⁺ cells in colon tissues from DSS-challenged DP1^{F/F}Tie2^{Cre} mice. Indeed, we observed 22% reduction of DP1 expression in colonic CD11b⁺F4/80⁺ cells from DP1^{F/F}Tie2^{Cre} mice compared to DP1^{F/F} controls (**Reviewer Figure 2**; [UNPUBLISHED DATA REMOVED PER AUTHOR REQUEST]). However, overall severity of both DSS- and TNBS-induced colitis in DP1^{F/F}Tie2^{Cre} mice was not higher than that in DP1^{F/F}LysM^{Cre} mice, including body weight loss, DAI and colon length (**Figure 4D-4F, Figure EV5**). So, we mentioned in the results and cited the relevant reference, please see **Page 8, Line 14**.

4. The cause-effect relationships of vascular permeability, epithelial apoptosis, and inflammatory signaling in myeloid cells has not been addressed. How do these alterations contribute to the pathology in a DP1-dependent manner? For example, is vascular permeability, and epithelial apoptosis also altered in LysM-Cre mice because they have less inflammatory mediators and, vice versa, how is the vascular permeability and myeloid inflammatory response altered in Villin-Cre mice?

This is a really good question. UC and Crohn's disease are chronic inflammatory bowel diseases, the exact etiology is unknown. Indeed, there must be an interrelationship among increased vascular permeability, epithelial apoptosis, and proinflammatory signaling in macrophages. For instance, epithelial cell apoptosis induces epithelial tight junction dysfunction leading to immune activation in experimental colitis [*Gastroenterology* 2009; 136, 551-563.]; In response to TNF α , macrophages produce reactive nitrosative species and results in epithelial apoptosis in animal models of IBD [*Gut.* 1995;37:247-255.]; Increased colonic vascular permeability causes epithelial hypoxia and apoptosis during the development of ulcerative colitis [*Curr Pharm Des* 2013;19:17-28]. As such, more severe epithelium loss and inflammatory reaction were observed in experimental colitis in epithelium- and myeloid- DP1 deficient mice (DP1^{F/F}Villin^{Cre} and DP1^{F/F}LysM^{Cre}, see **Figure 6 and Figure 7**), while severe lamina propria edema and epithelium loss in EC-DP1 deficiency mice (perhaps also in hematopoietic cells, DP1^{F/F}Tie2^{Cre}, see **Figure 5**). Therefore, we mentioned in the Introduction (**Page 3, Lines, 7-10**), modified the Results (**Page 8, Line 14**) and expanded the discussion (**Page 13, Line 8-9**),

5. Figure 2F: The time window is too short to monitor an effect of niacin on survival in WT mice.

Thanks, we corrected and repeated, please see **new Figure 2F**.

6. Figure 6A: Quantification for these surface markers would be more reliable if done by flow cytometry.

We did, please see **Figure 7B**.

7. There is no placebo group included in the clinical study. This is a major omission.

We appreciated the reviewer for the important concern. This is an explorative study with small sample size. We chose the patients with moderately active UC limited only in left side, which had no response to conventional retention enema treatment and regular oral medicines. These patients are suitable for retention enema therapy. We applied patent for it and are ready for larger scale clinical study. Thus, we discussed this limitation, please see **Page 14, last Paragraph**.

2nd Editorial Decision

07 February 2017

Thank you for the submission of your revised manuscript to EMBO Molecular Medicine. We have now received the enclosed reports from the referees that were asked to re-assess it. As you will see the reviewers are now globally supportive and I am pleased to inform you that we will be able to accept your manuscript pending the following final amendments:

1) Please address the minor text change commented by referee 2. Please provide a letter INCLUDING the reviewer's reports and your detailed responses to their comments (as Word file).

Please submit your revised manuscript within two weeks. May I suggest that you pay particular

attention to the right formatting at this stage; if the provided files are not in a publishable form, this will delay publication. I look forward to seeing a revised form of your manuscript.

***** Reviewer's comments *****

Referee #2 (Comments on Novelty/Model System):

This study is interesting with a potential impact on IBD patient treatment. The experimental models used were appropriate, and they utilized state of the art technology and strategies to address the scientific questions. The authors also translated the animal findings to patients.

Referee #2 (Remarks):

The revised version is improved. The authors appropriately responded to the reviewer comments/concerns and performed most of the suggested additional experiments/assessments.

Referee #3 (Remarks):

The manuscript has been adequately revised. The authors have reformulated critical passages of the manuscript and have added new experiments and data. This has clearly strengthen the study. Some minor concerns/suggestions remain which should be addressed by the authors:

1. A more accurate title would be "Niacin ameliorates ulcerative colitis via prostaglandin D2 mediated D prostanoid receptor 1 activation".
2. In contrast to what the authors say in the rebuttal letter, Figure 2F has not been changed. It should be expanded to at least 20 days or to the time point when all animals have died or were euthanized.

2nd Revision - authors' response

15 February 2017

Editors' Summary:

1) Please address the minor text change commented by referee 3. Please provide a letter INCLUDING the reviewer's reports and your detailed responses to their comments (as Word file).

As requested by Referee 3, we modified the Text, see below Responses.

Referee #1 (Comments on Novelty/Model System):

This study is interesting with a potential impact on IBD patient treatment. The experimental models used were appropriate, and they utilized state of the art technology and strategies to address the scientific questions. The authors also translated the animal findings to patients.

Thank you very much.

Referee #2 (Remarks):

The revised version is improved. The authors appropriately responded to the reviewer comments/concerns and performed most of the suggested additional experiments/assessments.

Thank you very much.

Referee #3 (Remarks):

The manuscript has been adequately revised. The authors have reformulated critical passages of the manuscript and have added new experiments and data. This has clearly strengthen the study. Some minor concerns/suggestions remain which should be addressed by the authors:

1. A more accurate title would be "Niacin ameliorates ulcerative colitis via prostaglandin D2 mediated D prostanoid receptor 1 activation".

Thank you so much for the helpful suggestion, we corrected.

2. In contrast to what the authors say in the rebuttal letter, Figure 2F has not been changed. It should be expanded to at least 20 days or to the time point when all animals have died or were euthanized.

I apologize for the typo and confusion.

i) In the first-round revision, we amended a new Figure 1 (Niacin induces PGD₂ secretion in mice) as Reviewers suggested. So the original 'Figure 2F' should become new **Figure 3F** now. Indeed, the Figure 2F (without any changes) also presents survival rates of DSS-challenged DP1^{-/-} and DP2^{-/-} mice within 10 days.

ii) In the original version, we did not observe significant effects of niacin treatment on survival in mice (P=0.067, n=10). The reviewer suggested that the time window is too short to monitor an effect of niacin on survival. As requested, we repeated by adding more animals (n=17-20). Since most of DP1^{-/-} mice (~75%-80%, **Figure 2F** and **Figure 3F**) died within 10 days after DSS challenge, and the statistically significant difference between niacin and vehicle treatment was reached after expanding sample sizes (p=0.0205, n=17-20, see **Table S6**), so we did not extend extra time.

iii) As such, we did replace with new Figure 3, change p value and animal numbers, see **Figure legend Page 35, Lines 15-16**.

Corresponding Author Name: Ying Yu

Manuscript Number: EMM-2016-06987